# Genetic effects on the timing of parturition and links to fetal birth weight

The timing of parturition is crucial for neonatal survival and infant health. Yet, its genetic basis remains largely unresolved. We present a maternal genome-wide meta-analysis of gestational duration ($n$ = 195,555), identifying 22 associated loci (24 independent variants) and an enrichment in genes differentially expressed during labor. A meta-analysis of preterm delivery (18,797 cases, 260,246 controls) revealed seven associated loci and large genetic similarities with gestational duration. Analysis of the parental transmitted and nontransmitted alleles ($n$ = 136,833) shows that 15 of the gestational duration genetic variants act through the maternal genome, whereas 7 act both through the maternal and fetal genomes and 2 act only via the fetal genome. Finally, the maternal effects on gestational duration show signs of antagonistic pleiotropy with the fetal effects on birth weight: maternal alleles that increase gestational duration have negative fetal effects on birth weight. The present study provides insights into the genetic effects on the timing of parturition and the complex maternal–fetal relationship between gestational duration and birth weight.

In humans, similar to mammals broadly, the timing of delivery is crucial for neonatal survival and health. Preterm delivery is the world-leading direct cause of death in neonates and children under five years of age[1]. Although the rate of neonatal mortality has substantially decreased in recent years, the reduction attributable to preterm delivery is one of the lowest among the major causes of mortality[2]. This fact partly reflects the relatively poor knowledge of the processes governing the timing of delivery in humans. Parturition may be initiated by a diversity of biological and mechanical pathways. Some of these are part of the physiological timing process, whereas others may override pregnancy maintenance with fail-safe mechanisms (for example, in the case of uterine infection)[3]. The diversity of the mechanisms has led to the conceptualization of preterm delivery as a syndrome[4], with various pathophysiological processes contributing to its etiology. Both maternal and fetal genomes are involved in these mechanisms. Yet, genetic studies have identified only a handful of loci associated with the timing of parturition[5,6].

Gestational duration is the major determinant of birth weight (that is, the longer the gestation, the heavier the newborn). At the same time, uterine load is one of the known triggers of parturition[7], evidenced by half of twin pregnancies delivering preterm[8]. Both the maternal

and fetal genomes contribute to birth weight as well, as revealed in recent genome-wide association studies (GWAS)[9,10], and over evolutionary time may have even conflicted on gestational duration and birth weight, as proposed in the hypothesis of the genetic conflicts of pregnancy[11]. This hypothesis suggests that the maternal genome favors slightly shorter gestations and lower birth weight, whereas the fetal genome favors the opposite. Coadaptation theory, instead, suggests that maternal and fetal genomes may invest resources to achieve an optimal gestational duration or birth weight that increases fitness[12]. These known contributions, potential conflicts and coadaptation of gestational duration and birth weight may ultimately create a complex relationship between the two.

What and how distinct are the maternal genetic effects on gestational duration and preterm delivery? What is the relationship between fetal growth and gestational duration? Is there evidence suggesting maternal–fetal coadaptation on these traits? To address these questions, we conducted a GWAS meta-analysis of gestational duration and preterm and post-term delivery in >190,000 maternal samples with spontaneous onset of delivery. We further analyzed these results using the parental transmitted and nontransmitted alleles in >135,000 parent-offsprings.

✉e-mail: pol.sole.navais@gu.se; bo.jacobsson@obgyn.gu.se

## Results

### Genome-wide association analyses

We conducted a GWAS meta-analysis of gestational duration in 195,555 women of recent European ancestry (Supplementary Table 1), a fourfold increase in sample size compared to the largest published maternal GWAS of gestational duration to date[5]. After quality control (QC), genetic variants at 22 loci were associated with gestational duration at genome-wide significance (Fig. 1, Supplementary Table 2 and Supplementary Fig. 1). Approximate conditional and joint (COJO) analysis revealed two conditionally independent signals at *EBF1* and *KCNAB1* gene regions. Sixteen of the loci did not overlap with any previously reported gestational duration-associated locus[5]. Effect sizes were relatively small, ranging from 7 (*HIVEP3/EDN2*) to 27 (*MRPS22*) hours of gestation per allele (average duration of gestation = 282 days, 40.3 weeks). Heterogeneity in the effect estimates was limited to loci previously identified (*EBF1*, *WNT4*, *ADCY5*, *EEFSEC* and *AGTR2*), likely due to winner's curse[13] (Supplementary Table 2 and Supplementary Fig. 2). Out-of-sample reanalysis of previously reported gestational duration-associated lead single-nucleotide polymorphisms (SNPs) (*n* = 6) showed that all four that were available after QC replicate at nominal significance (Supplementary Table 3). In addition, all six loci (±250 kb from lead SNP) replicated at suggestive evidence.

To prioritize candidate genes, we performed colocalization analysis[14] with *cis*-expression quantitative trait loci (*cis*-eQTLs) in induced pluripotent stem cells[15], endometrium[16], uterus, vagina and ovary[17] (Supplementary Table 4). *cis*-eQTLs for seven protein-coding (*OPRL1*, *ZBTB38*, *RGS19*, *TET3*, *COL27A1*, *CRISPLD1* and *ADCY5*) and four non-coding genes colocalized with gestational duration. Furthermore, colocalization analysis with blood protein QTLs[18] showed several *trans* associations: *ZBTB38* with three proteins, and *TCEA2/OPRL1* and *WNT4* with one each. Particularly interesting are the associations with OPRL1 and POMC, which play a role in modulating nociception and pain perception; in vitro studies in tissues from pregnant rats and humans suggest that the administration of nociceptin inhibits uterine contractions, mediated by the OPRL1 receptors[19,20].

RNA tissue-specific enrichment of top genes highlighted the endometrium and other female reproductive and smooth muscle tissues (Supplementary Fig. 3), results further supported at the genome-wide scale using stratified linkage disequilibrium (LD)-score regression (Supplementary Fig. 4). Previous genetic studies have suggested a critical role of the decidua (endometrium) in the timing of parturition, indicating an effect early in pregnancy[21]. Using stratified LD-score regression, we show that the heritability of gestational duration is enriched in regions harboring genes differentially expressed during labor (enrichment = 1.7, *P* = 7.1 × 10⁻⁷; Extended Data Fig. 1)[22], suggesting the SNPs associated with gestational duration may as well act during labor.

Stratified LD-score regression (Supplementary Fig. 5) revealed an enrichment in background selection, superenhancers, CpG content, H3K23ac and DNA methylation. Using the mosaic pipeline[23], we confirm that gestational duration loci have diverse evolutionary histories, including evolutionary conservation, excess population differentiation and negative selection (Supplementary Fig. 6).

We also performed a GWAS meta-analyses of preterm delivery (controls, delivery between 39 and 42 gestational weeks, *n* = 260,246; cases, delivery <37 completed weeks, *n* = 18,797) and post-term delivery (controls, delivery between 39 and 42 gestational weeks = 115,307, cases >42 completed weeks, *n* = 15,972) (Fig. 1a, Supplementary Table 2 and Supplementary Figs. 7 and 8). We observed a lower number of associated loci: seven and one for preterm and post-term delivery, respectively. COJO analysis identified a secondary conditionally independent SNP associated with preterm delivery at the *EBF1* gene region. We identified only one locus associated with preterm delivery (rs312777, *P* = 6.6 × 10⁻⁹) that showed weak evidence of association with gestational duration (*P* = 3.9 × 10⁻³).

We observed a modest genetic correlation (*r*_g = −0.62; 95% confidence interval (CI) = −0.72, −0.51) between gestational duration and preterm delivery, suggesting similarities between the two phenotypes (Supplementary Figs. 9 and 10). Post-term delivery, instead, showed a perfect genetic correlation with gestational duration (*r*_g = 1.17; 95% CI = 0.93, 1.41), suggesting no differences in the maternal genetic effects on such traits.

### Resolving maternal–fetal effect origin

The genetic effects on pregnancy traits may be driven by two correlated genomes: the maternal and the fetal. To investigate whether the gestational duration signals originate in either or both genomes, we used phased genotype data to estimate the effects of the parental transmitted and nontransmitted alleles from 136,833 parent-offspring trios or mother-child duos (Fig. 1b, Supplementary Table 5 and Extended Data Fig. 2; the maternal samples of these duos/trios were part of the GWAS meta-analysis). Based on pattern similarity using Gaussian mixture model-based clustering[10], SNPs were assigned to three large groups. Of the 24 index variants, 15 had the highest probability of a maternal effect, seven of both maternal and fetal effects (five with opposite effect directions, and the remaining two with the same direction), and two were grouped as having a fetal-only effect: the first, independent of the parent of origin (*TFAP4*, probability = 0.57), and the second limited to the maternal transmitted allele (*EEFSEC*). Caution should be taken when interpreting the latter considering the low probability (0.47).

The index SNP at the *ADCY5* locus (rs28654158) had both maternal and fetal effects on gestational duration with the same effect direction. Interestingly, a SNP also located in the first intron of *ADCY5* harbors maternal and fetal effects on birth weight, but in opposite directions, attributed to the fetal insulin hypothesis[9,10]. The two index SNPs for gestational duration (rs28654158) and birth weight (rs11708067) are located 50 kb apart from each other and are in low LD (*r*² < 0.2). The birth weight SNP, also implicated in diabetes, likely acts through *ADCY5* (ref. 24), but it is unknown whether the gestational duration variant also acts through the same gene, although it colocalizes with *ADCY5* gene expression in the uterus (Supplementary Table 4). Despite being physically close to each other, differences between the two loci are evident in the traits they colocalize with. The gestational duration locus also affects fat-mass-related traits, whereas the birth weight locus affects glucose-related ones (Extended Data Fig. 3).

The only fetal index SNP identified to date in a GWAS (rs7594852; minor allele frequency = 0.49; beta = 0.37 days; 95% CI = 0.22, 0.51)[6] clustered as having a fetal-only effect (Supplementary Table 5, probability = 1), independent of the parent of origin (beta paternal transmitted allele = −0.42, *P* = 2.7 × 10⁻⁶).

### Polygenic score of gestational duration and preterm delivery

We built polygenic scores for gestational duration and preterm delivery using the corresponding GWAS results in the MoBa cohort (including the X chromosome) using LDpred2 (ref. 25) and estimated their effect on both traits. The polygenic score for gestational duration explains 2.2% of its variance (beta = 0.22 days per *z*-score; 95% CI = 0.02, 0.03; *n* = 3,943). The lowest decile had a mean gestational duration of 278 days (95% CI = 278, 279), whereas the highest decile had a mean of 283 days (95% CI = 282, 284) (Fig. 2). The polygenic score was also statistically significantly associated with preterm delivery (Supplementary Table 6 and Supplementary Fig. 11; odds ratio = 0.994; 95% CI = 0.990, 0.997) with an area under the curve of 0.61 (95% CI = 0.55, 0.67). For comparison, a polygenic score for preterm delivery was built using the same samples as above. This polygenic score was also significantly associated with preterm delivery (Supplementary Table 6 and Supplementary Fig. 11; odds ratio = 1.005, 95% CI = 1.001, 1.009), with effect estimate similar to that obtained for the gestational duration polygenic score (after matching the direction). This reflects the genetic similarity between gestational duration and preterm delivery.

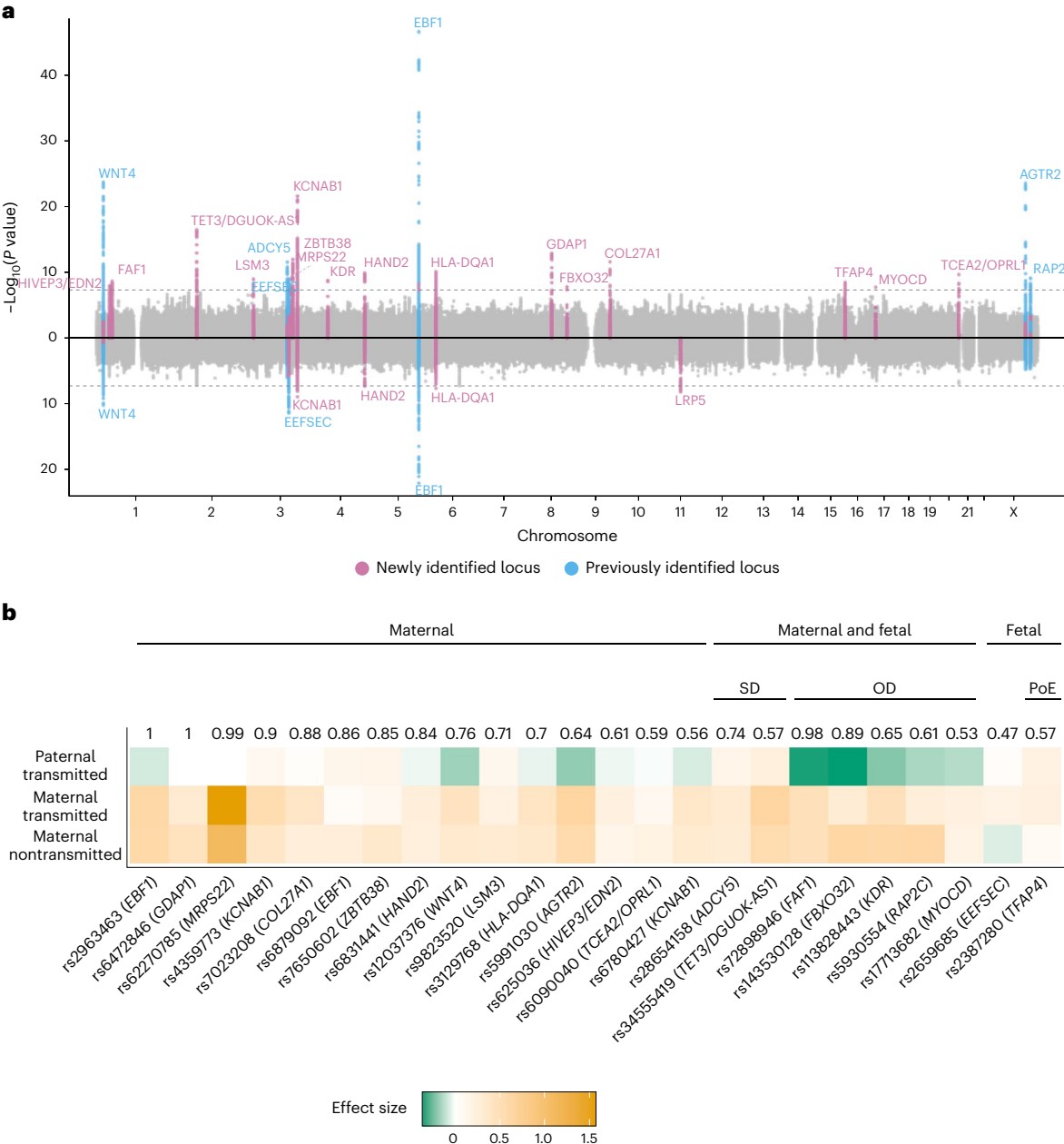

**Fig. 1 | GWAS of the timing of parturition and dissection of maternal–fetal effects. a**, Miami plot illustrating the GWAS for gestational duration (top) and preterm delivery (bottom). The *x*-axis shows the chromosome position and the *y*-axis the two-sided *P*-value of the fixed-effect inverse-variance weighted meta-analysis. The dashed line represents the genome-wide significance threshold ($P = 5 \times 10^{-8}$). Each genome-wide significant locus is labeled by its nearest protein-coding gene. **b**, Clustering of the effect origin for the index SNPs for gestational duration using transmitted and nontransmitted parental alleles ($n = 136,833$). Numbers depicted above the heatmap are the highest probability observed for that SNP, and group names define the cluster to which the highest probability

refers to. The probabilities were estimated using model-based clustering. Heatmap represents effect size and effect direction for the parental transmitted and nontransmitted alleles. For comparison purposes, the maternal alleles with positive effects were chosen as reference alleles. Three major groups were identified according to the highest probability: maternal-only effect, fetal-only effect and maternal and fetal effect. Within variants with both maternal and fetal effects, two clusters were observed: same (SD) or opposite (OD) effect direction from maternal and fetal genomes. One of the fetal effects was further clustered as having a parent-of-origin effect (PoE), specifically, an effect from the maternal transmitted allele.

## Pleiotropy between sex hormones and the timing of parturition

To examine the potential shared genetic basis between the timing of parturition and other traits, we estimated the genetic correlations between 14 female reproductive traits and the maternal effects on gestational duration and preterm delivery (Fig. 3). These estimates were generally comparable, with the latter being consistently higher. Calculated bioavailable testosterone (CBAT; $r_g = 0.40$; 95% CI = 0.26, 0.54), testosterone ($r_g = 0.35$; 95% CI = 0.19, 0.51) and sex hormone

binding globulin (SHBG; $r_g = -0.16$; 95% CI = -0.27, -0.06) in women were modestly genetically correlated with preterm delivery, whereas there was little genetic correlation with levels of the same hormones in men (Supplementary Table 7). We observed a positive genetic correlation between preterm delivery and the number of live births, and although this finding may be counterintuitive, it is in line with a positive genetic correlation reported between miscarriage and the number of live births[26]. The genetic correlation between preterm delivery and the

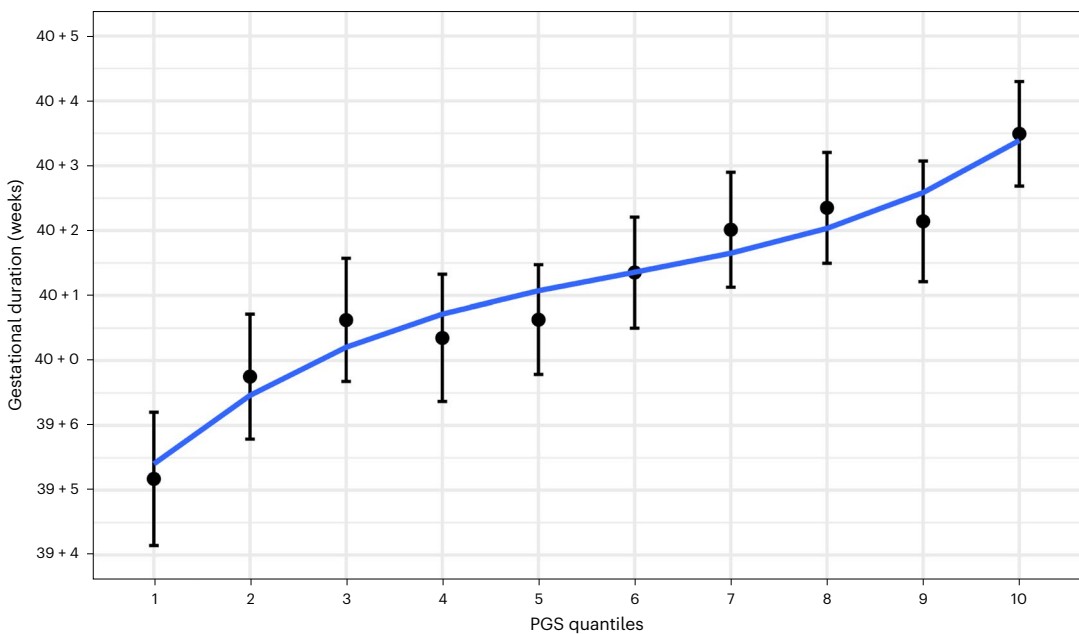

**Fig. 2 | Polygenic prediction of gestational duration.** Mean (95% CI) gestational duration for each decile of the gestational duration polygenic score ($n$ = 3,943). Only spontaneous deliveries were considered. PGS, polygenic score.

number of live births was twice as high in cohorts where the women's whole reproductive history was available ($r_g$ = 0.27; 95% CI = 0.11, 0.43) compared to cohorts based on a random pregnancy ($r_g$ = 0.13; 95% CI = 0.00, 0.26), indicating an increased probability of preterm delivery with an increasing number of live births. We also detected a negative genetic correlation with age at first birth and age at menopause.

Genetic correlations can arise due to pleiotropy or due to a trait being causally upstream of the other. To distinguish between these situations, we used a latent causal variable (LCV)[27] model between sex hormones and preterm delivery and gestational duration (Supplementary Table 8). We observed evidence for full or nearly full genetic causality of CBAT, testosterone and SHBG on preterm delivery (0.7 < GCP ≤ 0.8), but not on gestational duration (0.4 ≤ GCP < 0.5). In a two-sample Mendelian randomization analysis, the concentrations of these sex hormones (Supplementary Tables 9 and 10), including a set of variants that have consistent effects on testosterone, but no aggregate effects on SHBG[28], were associated with gestational duration and preterm delivery. Although the MR-Egger intercept was not significantly different from 0 (Supplementary Table 10 and Extended Data Fig. 4), colocalization analyses across the genome confirmed that distinct variants underlie the associations for sex hormones and the timing of parturition (Supplementary Fig. 12).

Using the parental transmitted and nontransmitted alleles in individual-level parent-offspring data from Iceland and Norway (deCODE, MoBa and HUNT; $n$ = 46,105 parent-offsprings; Supplementary Table 11), we observed a nominally significant association between the maternal nontransmitted alleles polygenic scores for CBAT and testosterone and gestational duration.

Testosterone and SHBG levels have a complex genetic link with the timing of parturition, likely explained by partial causality, as pointed out by the LCV analysis on gestational duration.

### Gestational duration partially mediates maternal effects on birth weight

We sought to understand the genetic relationship between gestational duration and birth weight and how the interplay between the maternal and fetal genomes affect this relationship. We used published summary statistics of birth weight (<15% of samples adjusted for gestational duration) derived from two different models[9]: maternal-only effect (adjusted by fetal effects) and fetal-only effect (adjusted by maternal effects). These models were obtained using weighted linear modeling and provide unbiased estimates for the maternal and fetal effects, respectively. The fetal effects on gestational duration were obtained from a previously published GWAS[6]. The more recent GWAS meta-analysis of fetal growth[10] had >40% of samples adjusted for gestational duration, which is the reason why we did not use it in this section.

The maternal effects on gestational duration are strongly correlated with those on birth weight (Supplementary Fig. 13; $r_g$ = 0.65; 95% CI = 0.54, 0.75). Conversely, neither the maternal ($r_g$ = −0.05; −0.15, 0.04) nor the fetal ($r_g$ = −0.02; 95% CI = −0.15, 0.11) effects on gestational duration were genetically correlated with the fetal-only effects on birth weight. We suggest the maternal effects on birth weight are at least partially mediated by gestational duration, whereas the effects of the fetus on birth weight are not.

We then tested the extent of this mediation. Using multitrait COJO analysis[29], we conditioned the genetic effects on birth weight on the maternal effects on gestational duration. After conditioning, the maternal effects on birth weight changed substantially: the SNP heritability was reduced by 53% ($P$ = 9.4 × $10^{-7}$; Supplementary Table 12), and the effect size of 87 suggestive SNPs decreased (Fig. 4a; median relative difference = −11%, Wilcoxon rank-sum test $P$ = 1.3 × $10^{-8}$). Applying the same method on genome-wide significant variants classified with a maternal-only effect on birth weight[9] provided very similar results (Supplementary Table 13 and Supplementary Fig. 14). This finding was further replicated using individual-level data by directly adjusting for gestational duration in the linear model on birth weight (using genotypes in Icelandic data and the maternal nontransmitted alleles in MoBa, Norway; Supplementary Table 13 and Supplementary Fig. 14). In contrast, for fetal effects on birth weight, conditioning on gestational duration did not change the effect estimates or the heritability (Fig. 4a and Supplementary Table 12 for results with 108 suggestive SNPs, and Supplementary Table 13 and Supplementary Fig. 14 with genome-wide significant variants classified as having a fetal-only effect[9]).

In summary, although the maternal effects on birth weight are partially driven by gestational duration, we found no evidence for this for the fetal effects on birth weight.

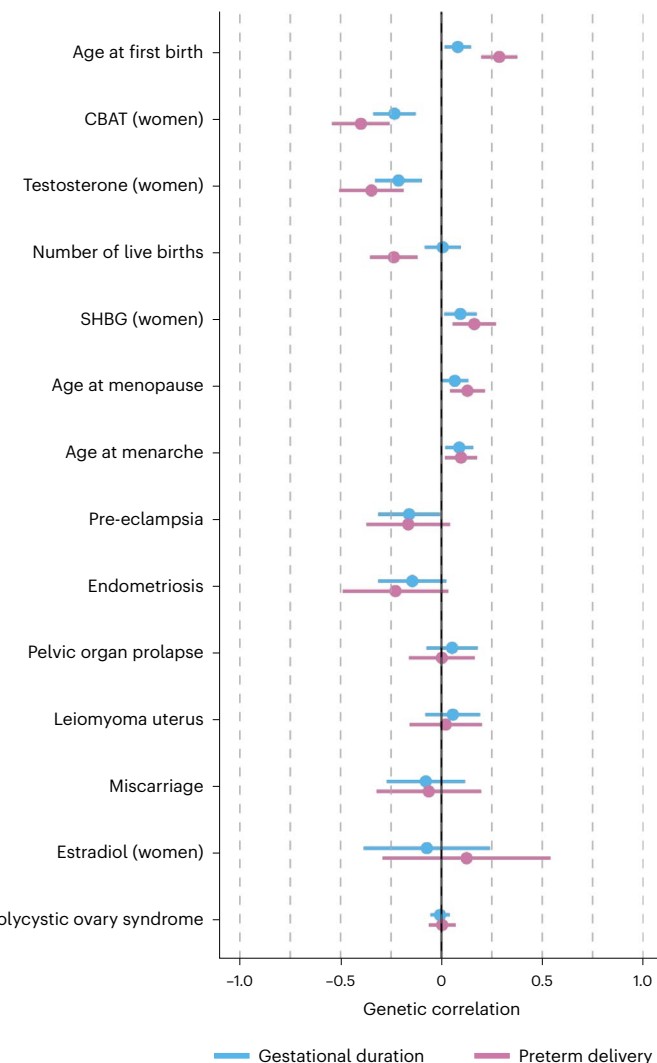

**Fig. 3 | Genetic correlations between gestational duration and preterm delivery and other female reproductive traits. a**, Genetic correlations between gestational duration ($n$ = 195,555) and preterm delivery (18,797 cases, 260,246 controls) and other female reproductive traits were estimated using LD-score regression. Dots are the genetic correlation estimate, and error bars are the 95% CI. The direction of the genetic correlations with preterm delivery was flipped so that term deliveries were considered as cases and preterm deliveries as controls. Hence, the direction of the genetic correlations of preterm delivery matches that of gestational duration, providing a clear comparison of the 95% CI.

## The maternal genome drives the association between gestational duration and birth weight

It is widely accepted that longer gestations lead to heavier newborns. Here, we sought to obtain causal estimates of the effect of gestational duration on birth weight.

We used the index SNPs from our discovery GWAS and the effect estimates from the maternal nontransmitted alleles as genetic instruments in a two-sample Mendelian randomization analysis (Fig. 4b and Supplementary Fig. 15) on the maternal-only effects on birth weight (derived using a weighted linear model[9]). The maternal nontransmitted gestational duration-increasing alleles were associated with higher birth weight (beta = 0.06 $z$-scores per day; 95% CI = 0.05, 0.08; $P$ = 1.7 × 10$^{-16}$). The estimated effect (approximately 23 g per day) is concordant with the phenotypic association between gestational duration and birth weight (25 g per day in 18,452 samples from the MoBa cohort). We observed no effect from the paternal transmitted gestational duration-increasing alleles on birth weight. The LCV model

confirmed a full or nearly full causal (GCP = 0.6, $P$ = 0.002; Supplementary Table 8) effect of gestational duration on birth weight.

## Maternal effects on gestational duration and fetal effects on birth weight exhibit signs of antagonistic pleiotropy

First, we evaluated the impact of fetal growth on gestational duration by instrumenting fetal growth using 68 SNPs with fetal-only effect on birth weight ($n$ = 35,280 and 48,741 parent-offsprings; Supplementary Table 14)[9]. Higher paternally transmitted birth weight score was associated with shorter duration of gestation, and the estimated effect was larger when estimated using the last menstrual period (beta = −1.9 days per $z$-score, $P$ = 4.0 × 10$^{-4}$) than ultrasound. This result supports previous evidence showing faster fetal growth is associated with shorter duration of gestation[30]. To investigate whether this was due to antagonistic pleiotropy between the fetal effects on birth weight and the maternal effects on gestational duration, we assessed the relation between birth weight-increasing alleles and maternal effects on gestational duration. The fetal birth weight-increasing alleles were not associated with maternal effects on gestational duration (Supplementary Table 15), suggesting that the results presented above are likely not due to antagonistic pleiotropic effects.

Next, we used summary statistics to investigate potential pleiotropy between the genetic effects on gestational duration and fetal birth weight. Using methods borrowed from Mendelian randomization analysis, we evaluated the association between the maternal gestational duration-increasing alleles and the fetal effects on birth weight. We observe that the alleles that increase gestational duration through a maternal effect tend to reduce birth weight through a fetal effect (Fig. 4c and Supplementary Table 15). Interestingly, this effect was not limited to the maternal transmitted alleles (beta = −0.02 $z$-scores per day; 95% CI = −0.03, −0.01; $P$ = 3.4 × 10$^{-4}$) but was also observed for the maternal nontransmitted gestational duration-increasing alleles (beta = −0.01 $z$-scores per day; 95% CI = −0.02, −0.01; $P$ = 6.2 × 10$^{-3}$). The paternal transmitted gestational duration-increasing alleles were not associated with fetal-only effects on birth weight (Supplementary Table 15).

## Discussion

The timing of parturition is crucial for neonatal survival and health. Yet, discovery of maternal and fetal genetic effects lags behind that of other pregnancy traits such as birth weight[9] and fetal growth[10]. In this GWAS meta-analysis of parturition timing, we identified 17 loci not previously reported, one of which was more strongly linked to preterm delivery than to gestational duration. The results support large similarities in the maternal genetic effects on gestational duration and preterm delivery. By including parent-offspring data with a similar sample size to that of the discovery GWAS, we were able to discern maternal from fetal effects with high certainty for most index SNPs. Finally, the results show a complex genetic relationship between the maternal and fetal genomes on gestational duration and birth weight.

Our understanding of the molecular signals governing the timing of parturition in humans has not advanced significantly. Previous genomic evidence suggests a critical role of the decidua[21], denoting an effect on the timing of parturition as early as implantation. We report that the SNP heritability of gestational duration is enriched in genes differentially expressed during labor in the myometrium. We suggest the maternal effects on the duration of gestation may as well act during labor, for instance, by inhibiting uterine contractions. Genetic studies of gestational duration may prove useful in the discovery of drug targets as tocolytic agents or for labor induction. At the same time, the genetic effects on gestational duration and preterm delivery are largely similar; this is opposed to the heterogeneity observed at the phenotypic and transcriptomic levels[31,32]. As an example, although the polygenic score of gestational duration is still inadequate for clinical use, it had a similar effect on preterm delivery as a polygenic score of preterm delivery itself.

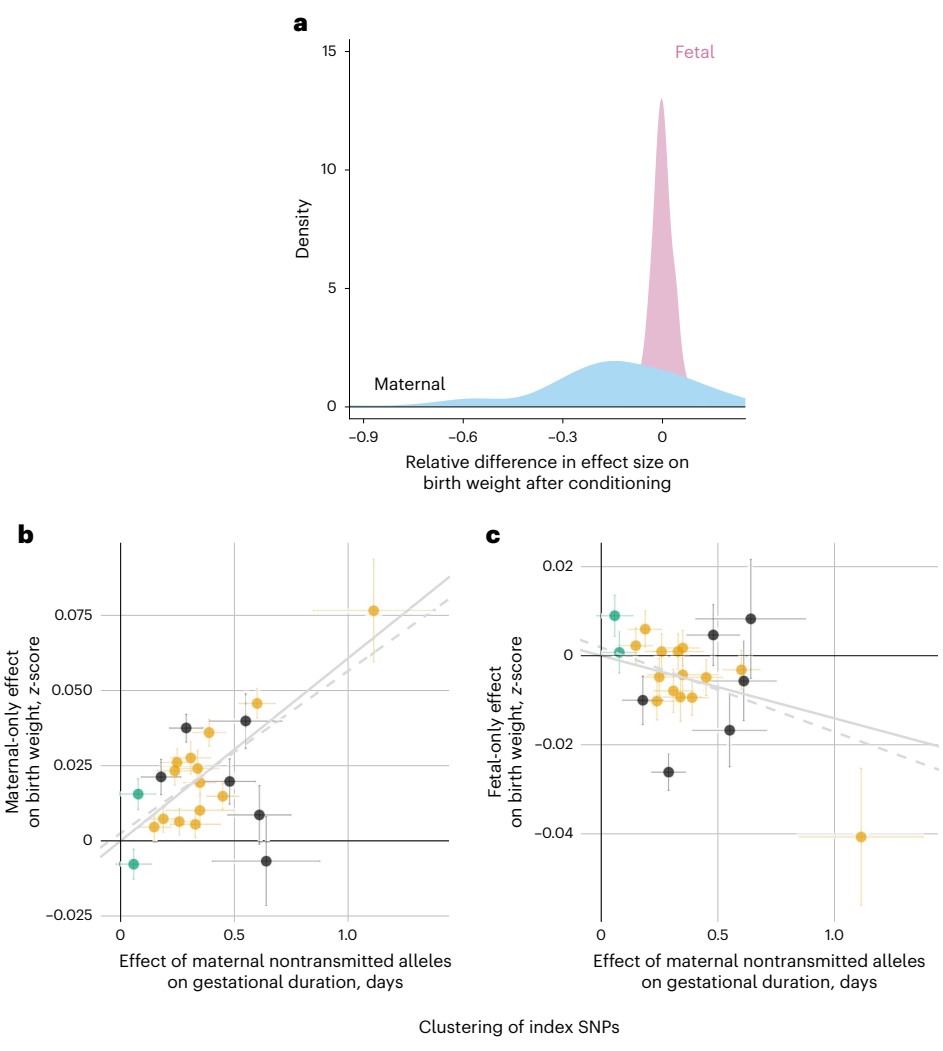

**Fig. 4 | Genetic relationship between gestational duration and birth weight. a**, Distribution of the relative difference in effect size before and after conditioning the effect on birth weight by the maternal effect on gestational duration using approximate multitrait COJO analysis. After conditioning, we split the genome into approximately LD-independent regions and selected the SNPs with the lowest *P* value on birth weight (*P* < 5 × 10⁻⁶) from each region (*n* SNPs maternal effect = 87; *n* SNPs fetal effect = 108). Fetal, pink; maternal, blue. **b,c**, Scatterplot for two-sample Mendelian randomization analysis for the maternal effect of gestational duration on birth weight (**b**, maternal effects; **c**, fetal effects). Each dot represents one of the gestational duration index SNPs. Effect sizes and standard errors (horizontal or vertical error bars) from the index SNPs for gestational duration derived from the maternal nontransmitted alleles were obtained from the meta-analysis of parent-offspring data (*n* = 136,833). The maternal-only and the fetal-only effects on birth weight were extracted from a previous GWAS meta-analysis (*n* = 210,248 and 297,356, respectively). The *x*-axis shows the SNP effect of the maternal nontransmitted alleles on gestational duration (days), and the *y*-axis the effect on birth weight (*z*-scores). Horizontal and vertical error bars represent the standard error. The solid line depicts the inverse-variance weighted method estimate, and the dashed line the MR-Egger estimate. Colors represent the clustering of the SNP effects on gestational duration, performed using model-based clustering.

Gestational duration is the major determinant of birth weight. Although the maternal genome affects offspring birth weight through many different causal pathways (for example, maternal glucose levels[9,10]), the effects are partly mediated by gestational duration. This has implications for the interpretation of GWAS of birth weight and downstream analyses, such as Mendelian randomization. In contrast, the fetal genetic effects on birth weight are not mediated by gestational duration, suggesting the fetal genome mainly acts on birth weight by modulating fetal growth. Interestingly, the maternal gestational duration-increasing alleles have negative fetal effects on birth weight, likely reflecting antagonistic pleiotropy. The opposite was not true; fetal birth weight-increasing alleles were not associated with maternal effects on gestational duration. We speculate that the fetal effects on birth weight have likely co-adapted to increase the fitness of the fetus in pregnancies genetically predisposed to a shorter duration. It has been suggested that both gestational duration and birth weight are under balancing selection, with intermediate values of these traits having highest fitness[3,33]. As exemplified here, this could lead to antagonistic pleiotropy favoring the coadaptation of maternal and fetal effects to attain optimal gestational duration and birth weight[12].

The presented results have several limitations. First, we analyzed data from participants of European ancestry. Over 70% of the samples were obtained from Nordic countries, with genotype data linked to the Medical Birth Registers; in these countries, the preterm delivery rate is one of the lowest in the world[1]. Studying diverse ancestries would propel the identification of novel loci associated with gestational duration and aid in fine-mapping efforts, as has been previously

shown for other traits[34]. Second, to understand the relationship between gestational duration and birth weight, we used summary statistics from a previously published birth weight GWAS that was partially adjusted for gestational duration (<15% of samples) and excluded preterm deliveries, which is likely to affect our analyses by reducing their power. Third, we assumed a causal association between gestational duration and birth weight. Although this is known to be true to some extent (that is, longer gestations are linked to heavier newborns), pleiotropy between gestational duration and birth weight could be very well at play. Fourth, phenotypic heterogeneity between cohorts (for example, gestational duration estimation method) may have hindered the identification of additional signals.

In conclusion, the present results provide evidence of large genetic similarities between gestational duration and preterm delivery and further our understanding of the complex relationship between gestational duration and birth weight. Particularly, we show that the maternal effects on birth weight are largely driven by gestational duration and that the maternal and fetal genomes have antagonistic pleiotropic effects on gestational duration and birth weight.

## Online content

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

Pol Solé-Navais [1] ✉, Christopher Flatley[1], Valgerdur Steinthorsdottir [2], Marc Vaudel [3], Julius Juodakis [1], Jing Chen [4,5], Triin Laisk [6], Abigail L. LaBella [7], David Westergaard [8,9,10], Jonas Bacelis [1], Ben Brumpton [11], Line Skotte [12], Maria C. Borges[13,14], Øyvind Helgeland [3,15], Anubha Mahajan [16,81], Matthias Wielscher [17,18], Frederick Lin [19], Catherine Briggs[20], Carol A. Wang [21,22], Gunn-Helen Moen [11,14,23,24], Robin N. Beaumont [25], Jonathan P. Bradfield[26], Abin Abraham[27], Gudmar Thorleifsson [2], Maiken E. Gabrielsen[11], Sisse R. Ostrowski [28,29], Dominika Modzelewska[1], Ellen A. Nohr[30], Elina Hyppönen [31,32], Amit Srivastava[5,33], Octavious Talbot[19], Catherine Allard [34], Scott M. Williams [35], Ramkumar Menon[36], Beverley M. Shields[25], Gardar Sveinbjornsson [2], Huan Xu[5,33], Mads Melbye[11,29,37,38], William Lowe Jr[19], Luigi Bouchard[39,40], Emily Oken[20], Ole B. Pedersen [29,41], Daniel F. Gudbjartsson [2,42], Christian Erikstrup [43,44], Erik Sørensen[28], Early Growth Genetics Consortium*, Estonian Biobank Research Team, Danish Blood Donor Study Genomic Consortium, Rolv T. Lie[37,45], Kari Teramo[46,47], Mikko Hallman[48,49], Thorhildur Juliusdottir[2], Hakon Hakonarson[50,51,52,53], Henrik Ullum[54], Andrew T. Hattersley [25], Line Sletner[55], Mario Merialdi[56], Sheryl L. Rifas-Shiman[20], Thora Steingrimsdottir[57,58], Denise Scholtens [19], Christine Power[59], Jane West [60], Mette Nyegaard [61], John A. Capra[62], Anne H. Skogholt [11], Per Magnus[37], Ole A. Andreassen [23,63,64], Unnur Thorsteinsdottir[2,57], Struan F. A. Grant [51,65,66,67], Elisabeth Qvigstad[23,68], Craig E. Pennell [21,22], Marie-France Hivert [20,69], Geoffrey M. Hayes [19], Marjo-Riitta Jarvelin [17,70,71,72], Mark I. McCarthy[16,81], Deborah A. Lawlor [13,14,73], Henriette S. Nielsen [9,29,74], Reedik Mägi [6], Antonis Rokas [7,75,76], Kristian Hveem[11,77,78], Kari Stefansson [2,57], Bjarke Feenstra [12], Pål Njolstad [3,79], Louis J. Muglia[5,33], Rachel M. Freathy [13,25], Stefan Johansson [3,80], Ge Zhang[5,33,82] & Bo Jacobsson [1,15,82] ✉

[1]Department of Obstetrics and Gynaecology, Sahlgrenska Academy, Institute of Clinical Science, University of Gothenburg, Gothenburg, Sweden. [2]deCODE genetics/Amgen, Reykjavik, Iceland. [3]Center for Diabetes Research, Department of Clinical Science, University of Bergen, Bergen, Norway. [4]Division of Biomedical Informatics, Cincinnati Children's Hospital Medical Center, Cincinnati, OH, USA. [5]Department of Pediatrics, University of Cincinnati College of Medicine, Cincinnati, OH, USA. [6]Estonian Genome Centre, Institute of Genomics, University of Tartu, Tartu, Estonia. [7]Department of Biological Sciences, Vanderbilt University, Nashville, TN, USA. [8]Novo Nordisk Foundation Center for Protein Research, University of Copenhagen, Copenhagen, Denmark. [9]Department of Obstetrics and Gynaecology, Copenhagen University Hospital Hvidovre, Hvidovre, Denmark. [10]Methods and Analysis, Statistics Denmark, Copenhagen, Denmark. [11]K.G. Jebsen Center for Genetic Epidemiology, Department of Public Health and Nursing, Norwegian University of Science and Technology, Trondheim, Norway. [12]Department of Epidemiology Research, Statens Serum Institut, Copenhagen, Denmark. [13]MRC Integrative Epidemiology Unit, University of Bristol, Bristol, UK. [14]Population Health Sciences, Bristol Medical School, University of Bristol, Bristol, UK. [15]Department of Genetics and Bioinformatics, Health Data and Digitalization, Norwegian Institute of Public Health, Oslo, Norway. [16]Wellcome Trust Centre for Human Genetics, University of Oxford, Oxford, UK. [17]Department of Epidemiology and Biostatistics, MRC-PHE Centre for Environment and Health, School of Public Health, Imperial College London, London, UK. [18]Department of Dermatology, Medical University of Vienna, Vienna, Austria. [19]Northwestern University Feinberg School of Medicine, Chicago, IL, USA. [20]Department of Population Medicine, Harvard Medical School, Harvard Pilgrim Health Care Institute, Boston, MA, USA. [21]School of Medicine and Public Health, College of Health, Medicine and Wellbeing, The University of Newcastle, Callaghan, New South Wales, Australia. [22]Hunter Medical Research Institute, New Lambton Heights, New South Wales, Australia. [23]Institute of Clinical Medicine, Faculty of Medicine, University of Oslo, Oslo, Norway. [24]University of Queensland Diamantina Institute, University of Queensland, Woolloongabba, Australia. [25]Institute of Biomedical and Clinical Science, College of Medicine and Health, University of Exeter, Exeter, UK. [26]Quantinuum Research LLC, Wayne, PA, USA. [27]Children's Hospital of Philadelphia, Philadelphia, PA, USA. [28]Department of Clinical Immunology, Copenhagen University Hospital - Rigshospitalet, Copenhagen, Denmark. [29]Department of Clinical Medicine, University of Copenhagen, Copenhagen, Denmark. [30]Research Unit of Gynecology and Obstetrics, Institute of Clinical Research, University of Southern Denmark, Odense, Denmark. [31]Australian Centre for Precision Health, Uni Clinical & Health Sciences, University of South Australia, Adelaide, Australia. [32]South Australian Health and Medical Research Institute, Adelaide, Australia. [33]Division of Human Genetics, Center for the Prevention of Preterm Birth, Perinatal Institute, Cincinnati Children's Hospital Medical Center, Cincinnati, OH, USA. [34]Centre de recherche du Centre hospitalier universitaire de Sherbrooke (CHUS), Sherbrooke, Québec, Canada. [35]Department of Population and Quantitative Health Sciences, School of Medicine, Case Western Reserve University, Cleveland, OH, USA. [36]Department of Obstetrics and Gynaecology, University of Texas Medical Branch, Galveston, TX, USA. [37]Centre for Fertility and Health, Norwegian Institute of Public Health, Oslo, Norway. [38]Department of Genetics, Stanford University School of Medicine, Stanford, CA, USA. [39]Department of Biochemistry and Functional Genomics, Faculty of Medicine and Health Sciences, Université de Sherbrooke, Sherbrooke, Québec, Canada. [40]Clinical Department of Laboratory Medicine, Centre intégré universitaire de santé et de services sociaux (CIUSSS) du Saguenay-Lac-St-Jean - Hôpital Universitaire de Chicoutimi, Saguenay, Québec, Canada.

[41]Department of Clinical Immunology, Zealand University Hospital, Køge, Denmark. [42]School of Engineering and Natural Sciences, University of Iceland, Reykjavik, Iceland. [43]Department of Clinical Immunology, Aarhus University Hospital, Aarhus, Denmark. [44]Department of Clinical Medicine, Faculty of Health, University of Aarhus, Aarhus, Denmark. [45]Department of Global Public Health and Primary Care, University of Bergen, Bergen, Norway. [46]Department of Obstetrics and Gynecology, Helsinki University Central Hospital, Helsinki, Finland. [47]University of Helsinki, Helsinki, Finland. [48]PEDEGO Research Unit and Medical Research Center Oulu, University of Oulu, Oulu, Finland. [49]Department of Children and Adolescents, Oulu University Hospital, Oulu, Finland. [50]The Center for Applied Genomics, Children's Hospital of Philadelphia, Philadelphia, PA, USA. [51]Department of Pediatrics, The Perelman School of Medicine, University of Pennsylvania, Philadelphia, PA, USA. [52]Division of Human Genetics, Children's Hospital of Philadelphia, Philadelphia, PA, USA. [53]Division of Pulmonary Medicine, Children's Hospital of Philadelphia, Philadelphia, PA, USA. [54]State Serum Institute, Copenhagen, Denmark. [55]Department of Pediatric and Adolescents Medicine, Akershus University Hospital, Lørenskog, Norway. [56]Maternal Newborn Health Innovations, PBC, Geneva, Switzerland. [57]Faculty of Medicine, University of Iceland, Reykjavik, Iceland. [58]Department of Obstetrics and Gynecology, Landspitali – The National University Hospital of Iceland, Reykjavik, Iceland. [59]Population, Policy, Practice. Great Ormond Street Institute of Child Health, University College London, London, UK. [60]Bradford Institute for Health Research, Bradford Teaching Hospitals NHS Foundation Trust, Bradford, UK. [61]Department of Health Science and Technology, Aalborg University, Aalborg, Denmark. [62]Bakar Computational Health Sciences Institute and Department of Epidemiology and Statistics, University of California San Francisco, San Francisco, CA, USA. [63]NORMENT Centre, University of Oslo, Oslo, Norway. [64]Division of Mental Health and Addiction, Oslo University Hospital, Oslo, Norway. [65]Center for Spatial and Functional Genomics Children's Hospital of Philadelphia, Philadelphia, PA, USA. [66]Divisions of Human Genetics and Endocrinology and Diabetes, Children's Hospital of Philadelphia, Philadelphia, PA, USA. [67]Department of Genetics, The Perelman School of Medicine, University of Pennsylvania, Philadelphia, PA, USA. [68]Department of Endocrinology, Morbid Obesity and Preventive Medicine, Oslo University Hospital, Oslo, Norway. [69]Diabetes Unit, Massachusetts General Hospital, Boston, MA, USA. [70]Center for Life Course Health Research, Faculty of Medicine, University of Oulu, Oulu, Finland. [71]Biocenter of Oulu, University of Oulu, Linnanmaa, Oulu, Finland. [72]Department of Life Sciences, College of Health and Life Sciences, Brunel University London, Uxbridge, UK. [73]NIHR Bristol Biomedical Research Centre, Bristol, UK. [74]The Recurrent Pregnancy Loss Unit, The Capital Region, Copenhagen University Hospitals Rigshospitalet & Hvidovre Hospital, Hvidovre, Denmark. [75]Department of Biomedical Informatics, Vanderbilt University School of Medicine, Nashville, TN, USA. [76]Vanderbilt Genetics Institute, Vanderbilt University, Nashville, TN, USA. [77]HUNT Research Centre, Department of Public Health and Nursing, Norwegian University of Science and Technology, Levanger, Norway. [78]Department of Medicine, Levanger Hospital, Nord-Trøndelag Hospital Trust, Levanger, Norway. [79]Children and Youth Clinic, Haukeland University Hospital, Bergen, Norway. [80]Department of Medical Genetics, Haukeland University Hospital, Bergen, Norway. [81]Present address: Genentech, South San Francisco, CA, USA. [82]These authors contributed equally: Ge Zhang, Bo Jacobsson. *Lists of authors and their affiliations appear at the end of the paper. ✉e-mail: pol.sole.navais@gu.se; bo.jacobsson@obgyn.gu.se

**Early Growth Genetics Consortium**

Pol Solé-Navais[1], Christopher Flatley[1], Marc Vaudel[3], Jonas Bacelis[1], Line Skotte[12], Øyvind Helgeland[3,15], Carol A. Wang[21,22], Gunn-Helen Moen[11,14,23,24], Robin N. Beaumont[25], Jonathan P. Bradfield[26], Ellen A. Nohr[30], Elina Hypponen[31,32], Mads Melbye[11,29,37,38], Emily Oken[20], Hakon Hakonarson[50,51,52,53], Christine Power[59], Per Magnus[37], Struan F. A. Grant[51,65,66,67], Craig E. Pennell[21,22], Marie-France Hivert[20,69], Geoffrey M. Hayes[19], Marjo-Riitta Jarvelin[17,70,71,72], Mark I. McCarthy[16], Deborah A. Lawlor[13,14,73], Reedik Mägi[6], Bjarke Feenstra[12], Pål Njolstad[3,79], Louis J. Muglia[5,33], Rachel M. Freathy[13,25], Stefan Johansson[3,80], Ge Zhang[5,33,82] & Bo Jacobsson[1,15,82]

**Estonian Biobank Research Team**

Triin Laisk[6] & Reedik Mägi[6]

**Danish Blood Donor Study Genomic Consortium**

Sisse R. Ostrowski[28,29], Ole B. Pedersen[29,41], Daniel F. Gudbjartsson[2,42], Christian Erikstrup[43,44], Erik Sørensen[28], Andrew T. Hattersley[25], Mette Nyegaard[61], Unnur Thorsteinsdottir[2,57] & Kari Stefansson[2,57]

## Methods

### Phenotype definition

In this study, we included pregnancies with a singleton live birth and a spontaneous onset of delivery: medically initiated deliveries (either by induction or planned cesarean section) were excluded or part of controls for preterm delivery. Gestational duration in days was estimated using either the last menstrual period date or ultrasound. We excluded pregnancies lasting <140 days (20 completed weeks) or >310 days (44 completed weeks), as well as women with health complications prior to or during pregnancy and congenital fetal malformations. Spontaneous preterm delivery was defined as a spontaneous delivery <259 days (37 completed gestational weeks) or by using the ICD-10 O60 code, and controls as a delivery occurring between 273 and 294 days (39 and 42 gestational weeks). Post-term delivery was defined as a delivery occurring >294 days (42 completed weeks) or ICD-10 O48 code, and controls as a spontaneous delivery between 273 and 294 days (39 and 42 gestational weeks). Given the perfect genetic correlation between gestational duration and post-term delivery GWAS, and the small power of the latter, all downstream analyses are focused on gestational duration and preterm delivery.

### Study cohorts and individual-level GWAS

This study consists of cohorts participating in the Early Growth Genetics (EGG) Consortium and the Norwegian Mother, Father and Child Cohort study (MoBa)[35], deCODE genetics[10], Trøndelag Health Study (HUNT)[36], Danish Blood Donor Study (DBDS)[37], the Estonian Genome Center of the University of Tartu (EGCUT)[38] and summary statistics from FinnGen[39] and from a previous GWAS of gestational duration and preterm delivery performed using 23andMe data[5]. A total of 18 different cohorts (Supplementary Table 1) provided GWAS data under an additive model for meta-analysis for the maternal genome, resulting in 195,555 samples for gestational duration, 276,218 samples for preterm delivery (n cases = 18,797) and 131,279 samples for post-term delivery (n cases = 15,972) of recent European ancestries (indicated by principal component analysis). For binary outcomes (preterm and post-term deliveries), only cohorts with an effective sample size >100 were included. Detailed description of the cohorts included can be found in the Supplementary Note. All study participants provided a signed informed consent, and all research studies were approved by the relevant institutional ethics review boards (Supplementary Note).

Each individual cohort applied specific QC procedures, data imputation and analysis independently following the consortium recommendations. Unless more stringent, samples were excluded if genotype call rate <95%, autosomal mean heterozygosity >3 standard deviations from the cohort mean, sex mismatch or major recent ancestry was other than European (HapMap central European). Genetic variants were excluded if genotype call rate <98%, Hardy-Weinberg equilibrium $P$ value < $1 \times 10^{-6}$ or minor allele frequency <1%. Reference panels for imputation were either 1000 Genomes Project[40], Haplotype Reference Consortium[41], 10KUK or a combination of one of the mentioned reference panels and own whole-genome sequencing data (deCODE, HUNT, DBDS and FinnGen). Each individual cohort performed a GWAS using an additive linear regression model adjusted for, at least, genetic principal components or relationship matrix on autosomal chromosomes and chromosome X. Summary statistics for each individual cohort were stored centrally and underwent QC procedures before meta-analysis (Supplementary Note).

### Meta-analysis of GWASs

After QC, individual-cohort GWAS summary statistics were pooled using fixed-effects inverse-variance weighted meta-analysis with METAL[42] without genomic control correction. We also performed an analysis of heterogeneity of effects (Supplementary Table 2; $I^2$ statistic). After meta-analysis, we removed genetic variants reported in less than half the number of available samples for each phenotype, resulting in 9-10 million genetic variants. For example, the variant observed in the largest number of samples for gestational duration was available in 195,555 individuals; only variants reported in at least 97,778 were kept. Genomic inflation factors were low for all three phenotypes (Supplementary Table 16; gestational duration $\lambda = 1.14$, preterm delivery $\lambda = 1.08$ and post-term delivery $\lambda = 1.05$). LD-score regression intercepts were substantially lower than genomic inflation factors, suggesting that the inflation in test statistics was mostly due to polygenicity (Supplementary Table 16). Test statistics were not further adjusted for genomic control for any of the phenotypes. If not otherwise stated, all analyses presented in this study are two-sided tests.

Initially, we naively defined independent loci based on physical distance, where SNPs within 250 kb from the index SNP were considered to be at the same locus. Novel loci were defined as loci not overlapping previously reported gestational duration loci in the largest GWAS performed to date[5]. Finally, we used conditional analysis to resolve independent loci (see below).

### Conditional analysis

We looked for conditionally independent associations within each locus using approximate conditional and joint (COJO) analysis[43] implemented in Genome-wide Complex Trait Analysis (GCTA) software[44]. We ran a stepwise model selection (-cojo-slct) to identify conditionally independent genetic variants at $P < 5 \times 10^{-8}$ for each of the genome-wide significant loci (using a radius of 1.5 Mb from the index SNP). Overlapping loci were merged into a single locus (only two loci overlapped, at 3q23). LD between genetic variants was estimated from 19,092 maternal samples from the Norwegian Mother, Father and Child Cohort, after excluding variants with imputation INFO score <0.4. We converted the reference panel from BGEN files to hard-called PLINK binary format (.bed). As per default in COJO, genetic variants >10 Mb apart were assumed to be in complete linkage equilibrium.

### Gene prioritization

To prioritize genes at the gestational duration loci identified, we set the baseline as the nearest protein-coding gene to the index SNP at each independent locus. Although naive, this approach has been consistently shown to outperform other single metrics for locus-to-gene mapping[45,46]. Next, we performed colocalization analysis for $cis$-eQTLs in 1,367 human induced pluripotent stem cell lines from the i2QTL resource (±250 kb from gene start and stop position)[15], endometrium (± 250 kb from gene start and stop position)[16] and uterus, vagina and ovary from GTEx (±1 Mb around transcription start site)[17]. None of the variants we identified were in LD ($r^2 > 0.6$) with missense variants. To complement the prioritization of genes, we queried each of the index SNPs for blood protein QTLs[18] (both in $cis$ and $trans$). For all index SNPs that were protein QTLs ($P < 5 \times 10^{-6}$), we performed colocalization analyses (±1.5 Mb around the index SNP). We excluded the $HLA$ region due to its large pleiotropic effects.

### Colocalization

We utilized genetic colocalization to identify pleiotropic effects between gestational duration and expression and protein quantitative trait locus (see Gene prioritization) and with other female and reproductive traits. To this end, we applied COLOC[14], which evaluates, in a Bayesian statistical framework, whether a single locus from two different phenotypes best fits a model where the associations are due to a single shared variant or distinct variants in close LD (Supplementary Note).

Prior probabilities for each for the non-null hypotheses were set as suggested by Wallace (prior probabilities that a random SNP in the loci is associated with phenotype A, phenotype B, or both phenotypes, $1 \times 10^{-4}$, $1 \times 10^{-4}$, and $5 \times 10^{-6}$, respectively), which are considered more conservative than the ones set by default[47]. Strong evidence of colocalization was defined as a posterior probability of colocalization >0.9.

## Enrichment analysis

We tested for enrichment based on top loci and genome-wide using partitioned LD-score regression. To test for overrepresentation in tissue-specific RNA expression (Human Protein Atlas, RNA consensus tissue gene data)[48], a Wilcoxon rank-sum test was performed on normalized RNA for genes within our set (above-mentioned) and all other genes. Significance for this test was set at Bonferroni correction for the number of tissues ($P < 0.05/61$), and suggestive evidence at $P < 0.1/61$. At the genome-wide level, we performed partitioned heritability using LD-score regression to test for enrichment in 97 different annotations[49,50], tissue-specific RNA expression using 205 different tissues/cell types[51], using precomputed partitioned LD-scores for subjects of recent European ancestry (baseline-LD model v2.2) and for enrichment in regions harboring genes differentially expressed during labor in single cells from myometrium[22].

## Genetic correlations

We estimated genetic correlations by performing LD-score regression[52] locally using precomputed LD-scores from 1000 Genomes Project samples of recent European ancestry. The MHC region (chr6:28477797-33448354) was removed prior to running LD-score regression.

## Resolving effect origin

To classify the identified index SNPs for gestational duration as having maternal, fetal, or maternal and fetal origin, we performed an association analysis using the parental transmitted and nontransmitted alleles on gestational duration. We used phased genotype data (that is estimated haplotypes) in parent-offsprings or mother-child duos to infer the parent-of-origin of the genotyped/imputed alleles as previously described[30]. Once the transmitted allele was identified, the nontransmitted maternal allele was extracted. Briefly, parental origin of each allele was inferred using genotypes of relatives, reference cohort data, or distributions of genotypes within the cohort and LD measurements. Different methods were used for phasing in each of the cohorts providing data for this analysis[10,53–56] (Supplementary Table 17). Details of the phasing strategy used by each cohort are described in Supplementary Note.

For each index SNP, we fit the following linear regression model:

$$\text{gestational duration} = MnT + MT + PT + PCs,$$

where $MnT$ and $MT$ refer to the maternal nontransmitted and transmitted alleles respectively, and $PT$ refers to the paternal transmitted alleles. The latter is interpreted as a fetal-only genetic effect, whereas the effect of the maternal nontransmitted allele is a maternal-only genetic effect. We first estimated the effects of the index SNPs in each birth cohort separately; effect sizes were then combined through fixed-effect meta-analysis, totaling a sample size of 136,833 (Supplementary Note and Supplementary Table 17). To classify the identified genetic variants into classes with similar patterns of effect, we used model-based clustering[10]. Variants were clustered based on estimated effects of the transmitted and nontransmitted parental alleles into five clusters. Two clusters assume fetal effect only, one with effect independent of parent of origin, and one where the effect is limited to the maternally transmitted allele; a cluster with maternal effect only; and two clusters with both maternal and fetal effects, either in opposite or same direction.

## Locus pleiotropy at 3q21

After identifying locus pleiotropy between the maternal effect on gestational duration and the fetal-only effect on birth weight at the *ADCY5* gene region, we set out to investigate differences between the two top SNPs in their colocalization with other traits. Phenome-wide colocalization for the two regions was performed using summary statistics from FinnGen (data freeze 5) and Pan UK Biobank data (https://pan.

ukbb.broadinstitute.org, in participants of recent European ancestry; Supplementary Note).

## Female reproductive traits

We obtained summary statistics for several female reproductive traits from different sources (minimum sample size 10,000). We included summary statistics from the following traits: miscarriage[26], gestational duration (fetal genome)[6], age at first birth, age at menarche (Neale lab, http://www.nealelab.is), age at menopause[57], number of live births (Neale lab, http://www.nealelab.is), testosterone[58], CBAT[58], SHBG[58], estradiol (women, Neale lab, http://www.nealelab.is), pelvic organ prolapse (FinnGen), polycystic ovary syndrome ([59] and FinnGen), endometriosis (Neale lab, http://www.nealelab.is), leiomyoma uterus (FinnGen) and pre-eclampsia[60]. For polycystic ovary syndrome, we meta-analyzed summary statistics from the largest published GWAS[59] and FinnGen. We estimated genetic correlations between gestational duration and preterm delivery and these traits, and latent causal variable analysis between sex hormones (testosterone, CBAT and SHBG) and gestational duration and preterm delivery. We further explored causality using two-sample Mendelian randomization and inspected whether the effects originated in the maternal or the fetal genome (see below, 'Mendelian randomization'). Finally, when one trait is causally upstream of the other, it is expected that the two traits would share a causal variant at some of the trait-associated loci. To test for this at the genome-wide scale, we performed colocalization analysis between sex hormones and gestational duration and preterm delivery using approximately LD-independent regions[61].

## Gestational duration and preterm delivery polygenic scores

To obtain an independent sample for training and validation of a polygenic score, the meta-analyses for gestational duration were rerun, excluding the MoBa cohort. These new meta-analysis results were used as the base data sets to calculate the polygenic scores. After applying the same exclusion criteria as used for the study samples in the meta-analysis, and removing duplicated samples and those with a kinship of greater than 0.125, the MoBa cohort was randomly split, using 80% ($n = 15,768$) as the training cohort and the remaining 20% ($n = 3,942$) as the validation cohort. LDpred2 was used for the calculation of the polygenic scores[25]. A description of polygenic score training can be found in Supplementary Note.

## Polygenic score validation

We constructed polygenic scores converted to $z$-scores to enable comparison of the gestational duration and the preterm delivery polygenic scores. To test the performance of the polygenic score, a linear regression was conducted for gestational duration by the polygenic score. A second model was used that adjusted for five principal components and genotyped batch. $R^2$ was calculated for the models to quantify variance explained.

The utility of the polygenic score for the prediction of preterm delivery was also assessed. Gestational duration was dichotomized into preterm delivery (<37 weeks) or full term (≥39 weeks and <41 weeks). Two models were analyzed, one assessing just the polygenic score and a second adjusting for five principal components and genotype batch. Receiver operating characteristic, area under the curve were calculated for each model and used as assessment of diagnostic accuracy.

## Mendelian randomization

We performed Mendelian randomization to study the effects of gestational duration (maternal) on birth weight (maternal) and the effects of fetal growth (fetal effect on birth weight) and sex hormones on gestational duration.

To study the effect of gestational duration on birth weight, we employed two-sample Mendelian randomization. The 24 index SNPs (22 autosomal SNPs) from the present gestational duration meta-analysis

and the effect sizes from the parental transmitted and nontransmitted alleles analysis were used to instrument gestational duration. Birth weight was instrumented using summary statistics from a previous GWAS of offspring's birth weight with minimal adjustment by gestational duration (<15% of samples)[9].

We assessed the effect of sex hormones (testosterone, SHBG and CBAT) on gestational duration using two-sample Mendelian randomization and instrumenting the hormones using a polygenic score for the parental transmitted and nontransmitted alleles. For each sex hormone, we obtained a list of independent SNPs genome-wide associated with these traits (Supplementary Table 9) by performing GWAS clumping ($r^2 > 0.001$) using the following PLINK command:

plink−bfile <1000 Genomes > −clump {GWAS summary statistics}−clump-r2 0.001−clump-kb 1000−clump-p1 5e-8−clump-p2 1e-5.

We also used a set of SNPs associated with testosterone, but with no aggregated effects on SHBG, as clustered in[28]. Such variants were used as instrumental variables in the two-sample Mendelian randomization analysis and to construct the polygenic score for the parental transmitted and nontransmitted alleles. The current meta-analysis results were employed as outcome for the two-sample Mendelian randomization analysis (inverse-variance weighted and MR-Egger). We subsequently constructed the polygenic score for the maternal transmitted and nontransmitted alleles and the paternal transmitted alleles in 46,105 parent-offsprings from Iceland and Norway. We estimated the effects of the maternal nontransmitted ($MnT_{PGS}$) and transmitted ($MT_{PGS}$) and paternal transmitted ($PT_{PGS}$) alleles polygenic score using the following linear model:

$$\text{gestational duration} = MnT_{PGS} + MT_{PGS} + PT_{PGS} + PCs + \text{batch}.$$

Again, effects from each of the three data sets (Iceland, MoBa and HUNT) were combined using fixed-effect inverse-variance weighted meta-analysis.

To understand the impact of fetal growth on gestational duration, we used individual genetic data from 35,280 (ultrasound-gestational duration) and 48,741 (last menstrual period-gestational duration) parent-offsprings from Iceland, the MoBa cohort and HUNT. To instrument fetal growth, we used 68 SNPs with fetal-only effect on birth weight as classified in Warrington et al. using Structural Equation Modeling[9]. Based on these 68 SNPs, we constructed a fetal growth polygenic score for the parental transmitted and nontransmitted alleles and regressed these on gestational duration (estimated by ultrasound or last menstrual period, separately). We estimated the effects of the maternal nontransmitted ($MnT_{PGS}$) and transmitted ($MT_{PGS}$) and paternal transmitted ($PT_{PGS}$) alleles polygenic scores as above.

Effect estimates from each of the three data sets (Iceland, MoBa and HUNT) were pooled using fixed-effects inverse-variance weighted meta-analysis.

### Multitrait conditional analysis

GCTA was used to perform bi-directional multitrait COJO (mtCOJO)[29] analysis using summary statistics. The gestational duration GWAS was conditioned on the birth weight GWAS and vice versa (Supplementary Note), using birth weight summary statistics from the largest GWAS meta-analysis of birth weight[9]. We did not condition on the fetal effects on gestational duration due to a lack of power in the fetal GWAS[6].

### Maternal–fetal pleiotropy on gestational duration and birth weight

We further investigated what are the fetal effects on birth weight for the maternal gestational duration-increasing alleles, and the maternal effects on gestational duration for the fetal birth weight-increasing alleles. To study this, we borrowed the inverse-variance weighted analysis from Mendelian randomization, but using the effects of two distinct genomes, the maternal and fetal. We caution that this should not be interpreted under a causal framework.

To understand what the maternal gestational duration-raising alleles do to birth weight when present in the fetus, we used the effect sizes and standard errors of the parental transmitted and nontransmitted alleles for the 22 autosomal index SNPs on gestational duration and assessed its effects on the same SNPs with a fetal-only effect on birth weight. To understand what the fetal birth weight-raising alleles do to gestational duration when present in the mother, we used the effect sizes and standard errors of 68 autosomal SNPs associated with fetal effects on birth weight and the effect sizes and standard errors from the current maternal GWAS of gestational duration.

### Evolutionary analysis

To examine the evolutionary history of the regions identified in the GWAS meta-analysis, we ran the significant variants through the MOSAIc pipeline[23]. This pipeline is designed to detect enrichment in evolutionary signals using a variety of sequence-based metrics of selection (Supplementary Note).

### Variant annotation

Variants were annotated using Ensembl's Variant Effect Predictor (hg19) command line tool[62]. Physical coordinates of protein-coding genes were obtained from the UCSC Table Browser[63], and were matched to the index SNPs using bedtools v2.29.2 (ref. [64]).

### Reporting summary

Further information on research design is available in the Nature Portfolio Reporting Summary linked to this article.

## Data availability

Cohorts should be contacted individually for access to raw genotype and phenotype data, as each cohort has different data access policies. Summary statistics from the meta-analysis, excluding 23andMe, are available at the EGG website (https://egg-consortium.org/), and access to the weights for constructing the polygenic score of gestational duration excluding 23andMe are available at the PGS Catalog (https://www.pgscatalog.org/, score ID: PGS002806). Access to the full set, including 23andMe results, can be obtained after approval from 23andMe is presented to the corresponding author or by completion of a Data Transfer Agreement (https://research.23andme.com/dataset-access/), which exists to protect the privacy of 23andMe participants. Access to the Danish National Birth Cohort (phs000103.v1.p1), Hyperglycemia and Adverse Pregnancy Outcome (phs000096.v4.p1) and Genomic and Proteomic Network (phs000714.v1.p1) individual-level phenotype and genetic data can be obtained through dbGaP Authorized Access portal (https://dbgap.ncbi.nlm.nih.gov/dbgap/aa/wga.cgi?page=login). The informed consent under which the data or samples were collected is the basis for determining the appropriateness of sharing data through unrestricted-access databases or NIH-designated controlled-access data repositories. The summary statistics used in this publication other than the one generated are available at the following links: fetal GWAS of gestational duration (http://egg-consortium.org/gestational-duration-2019.html), fetal and maternal GWAS of birth weight (http://egg-consortium.org/birth-weight-2019.html), miscarriage (http://www.geenivaramu.ee/tools/misc_sumstats.zip), age at first birth, estradiol (women), endometriosis, number of live births and age at menarche (http://www.nealelab.is), age at menopause (https://www.reprogen.org), testosterone (women)[58], SHBG, testosterone and CBAT (https://doi.org/10.6084/m9.figshare.c.5304500.v1), pelvic organ prolapse and leiomyoma of the uterus (https://www.finngen.fi/fi), polycystic ovary syndrome (https://www.repository.cam.ac.uk/handle/1810/283491 and https://www.finngen.fi/fi) and pre-eclampsia (European Genome-phenome Archive, https://ega-archive.org, EGAD00010001984). Pan-UK Biobank data are available at

https://pan.ukbb.broadinstitute.org/. Precomputed LD scores for European populations (https://data.broadinstitute.org/alkesgroup/LDSCORE/eur_w_ld_chr.tar.bz2) and multi-tissue gene expression precomputed stratified LD scores (https://alkesgroup.broadinstitute.org/LDSCORE/LDSC_SEG_ldscores/Multi_tissue_gene_expr_1000Gv3_ldscores.tgz) are available. eQTL data from GTEx are available at https://gtexportal.org/home/ and from endometrium at http://reproductivegenomics.com.au/shiny/endo_eqtl_rna/. Protein QTL data were obtained from https://www.omicscience.org/apps/pgwas/. Genome Reference Consortium Human Build 37 (hg19) available at https://www.ncbi.nlm.nih.gov/data-hub/genome/GCF_000001405.13/.

## Code availability
Code for this project has been structured using a Snakemake workflow[65] and is available at https://github.com/PerinatalLab/metaGWAS. A public release of it has been deposited in Zenodo (https://doi.org/10.5281/zenodo.7311977).

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

## Acknowledgements
B.J. received funding from The Swedish Research Council, Stockholm, Sweden (2015-02559 and 2019-01004), The Research Council of Norway, Oslo, Norway (FRIMEDBIO #547711, #273291) and March of Dimes (#21-FY16-121). Research reported in this publication (B.J., G.Z. and R.M.F.) was supported by the Eunice Kennedy Shriver National Institute Of Child Health & Human Development of the National Institutes of Health under Award Number R01HD101669. The content is solely the responsibility of the authors and does not necessarily represent the official views of the National Institutes of Health. G.-H.M. has received funding from the Norwegian Diabetes Association and Nils Normans minnegave. G.-H.M. is supported by the Norwegian Research Council (postdoctoral mobility research grant 287198). M.C.B.'s contribution to this work was supported by a UK Medical Research Council (MRC) Skills Development Fellowship (MR/P014054/1) and a University of Bristol Vice-Chancellor's fellowship. M.C.B. and D.A.L. are supported by the British Heart Foundation (AA/18/7/34219) and work in a Unit that receives funding from the University of Bristol and UK Medical Research Council (MRC) (MC_UU_00011/6). M.V. is supported by the Research Council of Norway (project #301178). D.A.L. is supported by a British Heart Foundation Chair (CH/F/20/90003). S.F.A.G. is supported by Daniel B. Burke Chair for Diabetes Research and NIH Grant R01 HD056465, IDF to CAG center from CHOP; CHOP's Endowed Chair in Genomic Research. Funding for T.L. was provided by the European Regional Development Fund and the programme Mobilitas Pluss (MOBTP155). B.M.S. is a core member of the NIHR Exeter Clinical Research Facility. R.M.F. and R.N.B. were funded by a Wellcome Trust and Royal Society Sir Henry Dale Fellowship (WT104150). R.M.F. is funded by a Wellcome Trust Senior Research Fellowship (WT220390). B.F. was supported by the

Oak Foundation. L.B. is a senior research scholar from the Fonds de la recherche du Québec en santé (FRQS) and member of the CR-CHUS, a FRQS-funded Research Center. E.O. has received funding from the US National Institutes of Health. D.W. is funded by the Novo Nordisk Foundation (NNF18SA0034956, NNF14CC0001, NNF17OC0027594). Additional funding statements for each cohort are available in the Supplementary Note.

## Author contributions

The core working group comprised P.S.-N., C.F., V.S., M.V., J.C., P.N., L.J.M., R.M.F., S.J., G.Z., B.J. The following authors performed analyses in their respective cohorts: P.S.-N., C.F., M.V., J.C., T.L., A.L.L., A.A., D.W., B.B., L.S., M.C.B., A.M., M.W., F.L., C.B., C.A.W., G.-H.M., R.N.B., J.P.B., G.T., O.T., G.S., H.X., D.F.G., S.L.R.-S., D.S. B.F. Individual cohort designers and principal investigators included V.S., J.B., J.J., A.M., M.E.G., E.A.N., E.H., S.M.W., R. Menon, M. Melbye, W.L., L.B., E.O., A.R., R.T.L., K.T., M.H., T.J., H.H., B.M.S., L.S., M. Merialdi, D.S., H.U., C.P., M.N., J.A.C., A.H.S., P.M., O.A.A., U.T., S.F.A.G., E.Q., C.E.P., M.-F.H., G.M.H., M.I.M., D.A.L., H.S.N., R. Mägi, K.H., K.S., B.F., L.J.M., R.M.F., S.J., G.Z. and B.J. Sample collection, phenotyping and/or genotyping was performed by V.S., J.B., J.J., Ø.H., C.A.W., G.T., M.E.G., S.R.O., D.M., E.A.N., E.H., A.S., C.A., A.T.H., M. Melbye, W.L., L.B., E.O., O.B.P., C.E., E.S., K.T., H.U., M.H., T.J., H.H., B.M.S., L.S., T.S., C.P., J.W., A.H.S., P.M., O.A.A., U.T., S.F.A.G., E.Q., C.E.P., M.-F.H., G.M.H., M.-R.J., M.I.M., D.A.L., R. Mägi, K.H., B.F., P.N., L.J.M., G.Z. and B.J.

## Funding

## Competing interests

As of January 2020, A.M. is an employee of Genentech and a holder of Roche stock. The views expressed in this article are those of the author(s) and not necessarily those of the NHS, NIHR or Department of Health. M.I.M. has served on advisory panels for Pfizer, Novo Nordisk and Zoe Global; has received honoraria from Merck, Pfizer, Novo Nordisk and Eli Lilly; and has received research funding from Abbvie, AstraZeneca, Boehringer Ingelheim, Eli Lilly, Janssen, Merck, Novo Nordisk, Pfizer, Roche, Sanofi Aventis, Servier and Takeda. As of June 2019, M.I.M. is an employee of Genentech and a holder of Roche stock. D.A.L. receives support from several national and international government and charitable research funders, as well as from Medtronic and Roche Diagnostics for research unrelated to that presented here. H.S. obtained speaker fees from Ferring Pharmaceuticals, Merck A/S, AstraZeneca and Cook Medical. V.S., G.T., G.S., D.F.G., U.T. and K.S. are employees of deCODE genetics/Amgen. The remaining authors declare no competing interests.

## Additional information

**Extended data** is available for this paper at

**Correspondence and requests for materials** should be addressed to
Pol Solé-Navais or Bo Jacobsson.

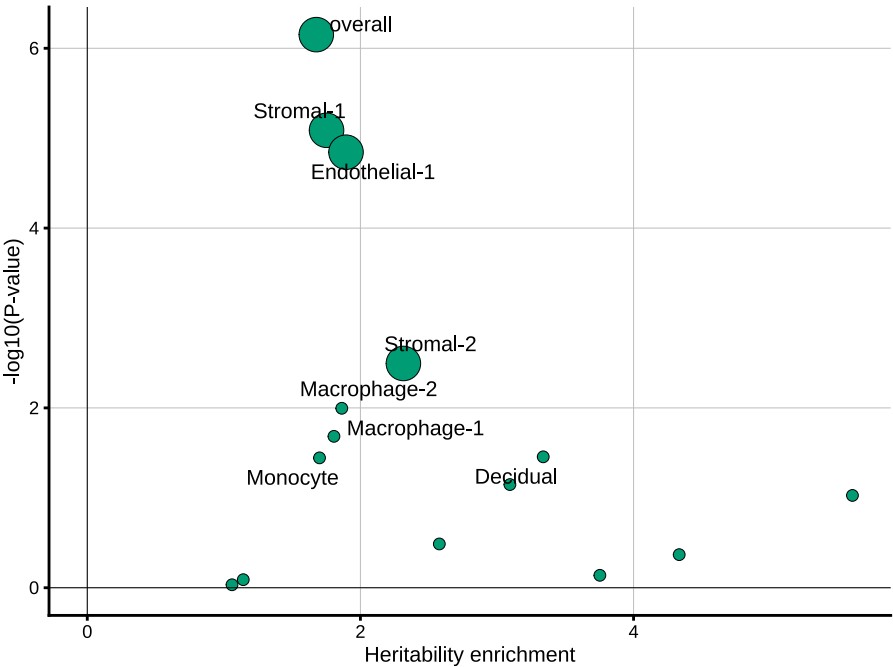

Significant after correction for the number of cell-types

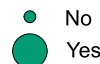

**Extended Data Fig. 1 | SNP-heritability enrichment of gestational duration for genes differentially expressed during labor in different cell types of the myometrium and overall.** LD-score regression was used to partition heritability and estimate the heritability enrichment for each cell type and overall. We calculated LD scores (European individuals from phase 3 of the 1000 Genomes project) for sets of genes differentially expressed at labor (±100 kb) for each cell type separately and for the overall set of genes differentially expressed in the myometrium. Each dot represents a cell type, the x-axis shows the heritability enrichment, and the y-axis the -log₁₀(P-value) of a two-sided test. Larger dots denote significant heritability enrichment after Bonferroni correction for multiple comparisons (that is, number of cell types; $P$-value < 0.05/15). See Online Methods for a cautionary note regarding the comparison of different cell-type enrichment $P$-values.

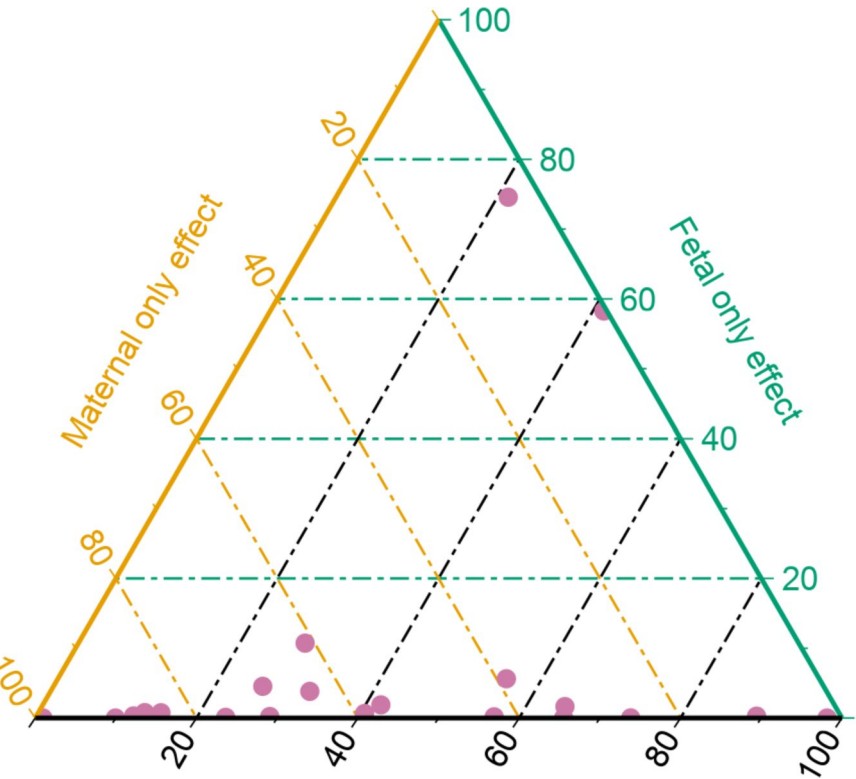

**Extended Data Fig. 2 | Ternary plot representing the probabilities of having maternal, fetal or maternal, and fetal effect for each index SNP.** The sum of all probabilities for each index SNP is 1. Lines are colored according to the axis they belong to. All points in a horizontal line (green) have the same probability of 'fetal-only effect', points on a line (yellow) parallel to the right side of the triangle have the same probability of a 'Maternal-only effect', and lines (black) parallel to the left side of the triangle have the same probability of a 'Maternal and fetal effect'. Probabilities were obtained using Gaussian Mixture models clustering using the effect size and standard error estimates of the parental transmitted and nontransmitted alleles ($n = 136,833$ parent-offsprings). While five different clusters were identified, the fetal effect was broken down into two groups (parent-of-origin and independent of parent-of-origin), and the maternal and fetal effects also into two groups (same or opposite maternal and fetal direction). For this figure, probability of a 'Fetal only effect' is the sum of the two groups with fetal effect, and 'Maternal and fetal effect' is the sum of the probabilities of the two clusters with maternal and fetal effects.

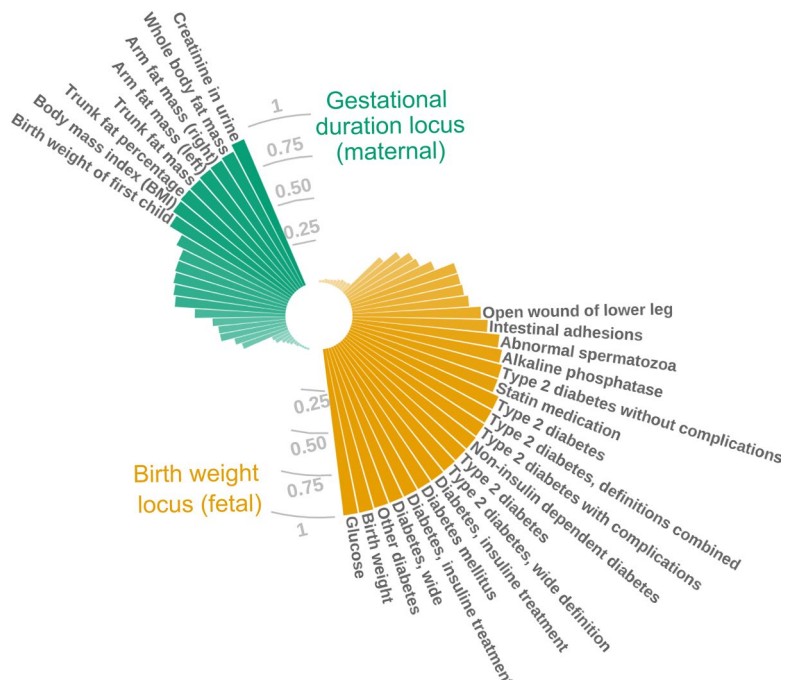

**Extended Data Fig. 3 | Colocalization between the maternal effects on gestational duration (green) and fetal effects on birth weight (yellow) and other phenotypes from UK Biobank and FinnGen at the *ADCY5* locus.** Posterior probability of colocalization between the maternal effect on gestational duration (rs28654158) and the fetal-only effect on birth weight (rs11708067) with traits from UK Biobank and FinnGen. Only traits with a posterior probability of colocalization ≥ 0.01 are plotted, and names are only shown if the posterior probability is > 0.5. Maternal locus on gestational duration was centered around rs28654158 (±1.5 Mb), and the fetal locus on birth weight around rs11708067 (±1.5 Mb).

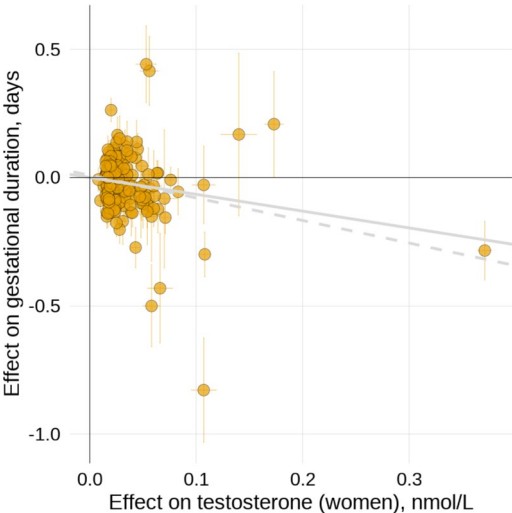

**Extended Data Fig. 4 | Association between testosterone levels (women) and maternal effect on gestational duration.** Scatterplot for two-sample Mendelian randomization analysis for the effect of testosterone in nmol/L (x-axis, independent of SHBG, $n$ = 230,454) on gestational duration in days (y-axis, maternal effect, $n$ = 195,555). Each dot represents one of the testosterone associated SNPs. Horizontal and vertical error bars represent the 95% CI. The gray line depicts the inverse-variance weighted method estimate, and the gray-dashed line the MR-Egger estimate.

# Reporting Summary

## Statistics

For all statistical analyses, confirm that the following items are present in the figure legend, table legend, main text, or Methods section.

| n/a | Confirmed | |
|---|---|---|
| ☐ | ☒ | The exact sample size (*n*) for each experimental group/condition, given as a discrete number and unit of measurement |
| ☐ | ☒ | A statement on whether measurements were taken from distinct samples or whether the same sample was measured repeatedly |
| ☐ | ☒ | The statistical test(s) used AND whether they are one- or two-sided<br>*Only common tests should be described solely by name; describe more complex techniques in the Methods section.* |
| ☐ | ☒ | A description of all covariates tested |
| ☐ | ☒ | A description of any assumptions or corrections, such as tests of normality and adjustment for multiple comparisons |
| ☐ | ☒ | A full description of the statistical parameters including central tendency (e.g. means) or other basic estimates (e.g. regression coefficient) AND variation (e.g. standard deviation) or associated estimates of uncertainty (e.g. confidence intervals) |
| ☐ | ☒ | For null hypothesis testing, the test statistic (e.g. $F$, $t$, $r$) with confidence intervals, effect sizes, degrees of freedom and $P$ value noted<br>*Give P values as exact values whenever suitable.* |
| ☐ | ☒ | For Bayesian analysis, information on the choice of priors and Markov chain Monte Carlo settings |
| ☐ | ☒ | For hierarchical and complex designs, identification of the appropriate level for tests and full reporting of outcomes |
| ☐ | ☒ | Estimates of effect sizes (e.g. Cohen's *d*, Pearson's *r*), indicating how they were calculated |

*Our web collection on statistics for biologists contains articles on many of the points above.*

## Software and code

Policy information about availability of computer code

| Data collection | No software was used for data collection. |
|---|---|
| Data analysis | Code for the meta-analyis and down-stream analyses can be found at https://github.com/PerinatalLab/metaGWAS.<br>The following software were used for data analysis: R version 4.1.1 (https://www.r-project.org) with packages: data.table 1.14.0, dplyr 1.0.7, tidyr 1.1.3, scales 1.1.1, knitr 1.33, cowplot 1.1.0, ggrepel 0.9.1, showtext 0.9_3, tidyverse 1.3.1, fmsb 0.7.1, ggtern 3.3.5, MendelianRandomization 0.5.1, gridextra 2.3, dendextend 1.15.1, plyr 1.8.6, ggtree 3.0.1, kableextra 1.3.4, metafor 4.3.0; coloc v3.0; Python version 3.7.9 with packages: pandas 1.1.4, numpy 1.19.5, urllib3 1.26.6, scipy 1.5.3; PLINK v1.90b6.6 64-bit (https://www.cog-genomics.org/plink/); PLINK v2.00a2.3LM (https://www.cog-genomics.org/plink/2.0/); METAL - version released on 2011-03-25 (https://genome.sph.umich.edu/wiki/METAL); bcftools 1.9 (https://samtools.github.io/bcftools/bcftools.html); ensembl-vep 106.1 (https://grch37.ensembl.org/Homo_sapiens/Tools/VEP); qctool v2.0.8 (https://www.well.ox.ac.uk/~gav/qctool/); GCTA 1.93.2beta (https://yanglab.westlake.edu.cn/software/gcta/#Overview); LCV (https://github.com/lukejoconnor/LCV, cloned the 24/08/2021); Python 2.7 with the following packages: scipy 0.18, pandas 0.20, numpy 1.16; LDSC v1.0.1 (https://github.com/bulik/ldsc, cloned the 27/05/2020); bedtools v2.29.2 (https://bedtools.readthedocs.io/en/latest/); BOLT-LMM v2.3 (https://alkesgroup.broadinstitute.org/BOLT-LMM/BOLT-LMM_manual.html#x1-5700012); LDpred2 (https://privefl.github.io/bigsnpr/articles/LDpred2.html); SHAPEIT2 (https://mathgen.stats.ox.ac.uk/genetics_software/shapeit/shapeit.html); EAGLE v2.3 (https://alkesgroup.broadinstitute.org/Eagle/#x1-40002.1) |

For manuscripts utilizing custom algorithms or software that are central to the research but not yet described in published literature, software must be made available to editors and reviewers. We strongly encourage code deposition in a community repository (e.g. GitHub). See the Nature Portfolio guidelines for submitting code & software for further information.

# Data

Policy information about <u>availability of data</u>

All manuscripts must include a <u>data availability statement</u>. This statement should provide the following information, where applicable:

- Accession codes, unique identifiers, or web links for publicly available datasets
- A description of any restrictions on data availability
- For clinical datasets or third party data, please ensure that the statement adheres to our <u>policy</u>

Cohorts should be contacted individually for access to raw genotype data, as each cohort has different data access policies. Summary statistics from the meta-analysis excluding 23andMe for each phenotype are available at the EGG website (egg-consortium.org/) and access to the weights for constructing the polygenic score of gestational duration excluding 23andMe are available at the PGS Catalog (https://www.pgscatalog.org/, score ID: PGS002806). Access to the full set, including 23andMe results, can be obtained after approval from 23andMe is presented to the corresponding author or by completion of a Data Transfer Agreement (https://research.23andme.com/dataset-access/), which exists to protect the privacy of 23andMe participants. Access to the Danish National Birth Cohort (phs000103.v1.p1), Hyperglycemia and Adverse Pregnancy Outcome (phs000096.v4.p1), and Genomic and Proteomic Network (phs000714.v1.p1) individual-level phenotype and genetic data can be obtained through dbGaP Authorized Access portal (https://dbgap.ncbi.nlm.nih.gov/dbgap/aa/wga.cgi?page=login). The informed consent under which the data or samples were collected is the basis for determining the appropriateness of sharing data through unrestricted-access databases or NIH-designated controlled-access data repositories. The summary statistics used in this publication other than the one generated are available at the following links: fetal GWAS of gestational duration (http://egg-consortium.org/gestational-duration-2019.html), fetal and maternal GWAS of birth weight (http://egg-consortium.org/birth-weight-2019.html), miscarriage (http://www.geenivaramu.ee/tools/misc_sumstats.zip), age at first birth, oestradiol (women), endometriosis, number of live births and age at menarche (http://www.nealelab.is), age at menopause (https://www.reprogen.org), testosterone (women) 59, SHBG, testosterone and CBAT (https://doi.org/10.6084/m9.figshare.c.5304500.v1), pelvic organ prolapse and leiomyoma of the uterus (https://www.finngen.fi/fi), polycystic ovary syndrome (https://www.repository.cam.ac.uk/handle/1810/283491 and https://www.finngen.fi/fi) and pre-eclampsia (European Genome-phenome Archive, https://ega-archive.org, EGAD00010001984). Pan-UK Biobank data is available at https://pan.ukbb.broadinstitute.org/. For pre-computed LD scores for European populations (https://data.broadinstitute.org/alkesgroup/LDSCORE/eur_w_ld_chr.tar.bz2), and for multi-tissue gene expression pre-computed stratified LD scores (https://alkesgroup.broadinstitute.org/LDSCORE/LDSC_SEG_ldscores/Multi_tissue_gene_expr_1000Gv3_ldscores.tgz). eQTL data from GTEx is available at https://gtexportal.org/home/ and from endometrium at http://reproductivegenomics.com.au/shiny/endo_eqtl_rna/. Protein QTL data was obtained from https://www.omicscience.org/apps/pgwas/. Genome Reference Consortium Human Build 37 (hg19) available at https://www.ncbi.nlm.nih.gov/data-hub/genome/GCF_000001405.13/.

# Human research participants

Policy information about <u>studies involving human research participants and Sex and Gender in Research.</u>

| | |
|---|---|
| Reporting on sex and gender | Findings from this work largely apply to one sex (women). The main GWAS meta-analysis was performed using only genotype and pregnancy information from women. In the analysis to distinguish maternal and fetal effects, we used parental data (including fathers, men) as well their offspring. Sex was determined using genotype data, and inconsistencies were excluded from the analysis. One section includes the use of a sex-stratified GWAS of adult horomone levels, due to large differences in the genetic effects on such hormones between sexes. |
| Population characteristics | The study included pregnant women of recent European ancestry with a singleton live-birth. Over 70% of the samples were recruited in Nordic countries (Iceland, Norway, Finland and Denmark), where the prevalence of spontaneous preterm delivery is 3%. Overall, the median preterm delivery rate was 9.2%, including case-control studies ascertained for preterm delivery. Post-term delivery rate was notably higher (approx. 13.1%) considering that no case-control studies were included. Median maternal age at birth ranged between 20-30 years old (median = 25), and nulliparous women (first pregnancy with delivery after 22 gestational weeks) were approximately half the sample size. |
| Recruitment | Recruitment and data collection was performed independently by each participating cohort, and is detailed in the Supplementary Note. Recruitment in cohorts participating in this GWAS meta-analysis can be largely grouped in three distinct categories: population-based, hospital-based or direct-to-consumer. Several cohorts were case-controls for preterm delivery, and have only been included as part of the preterm delivery GWAS meta-analysis. Most population-based cohorts originate from Nordic countries, where genetic data was linked to medical birth registers from each country. Only data from one direct-to-consumer genotyping company was employed (23andMe). Hospital-based cohorts were the most common, but overall accounted for a relatively small sample size. Participation bias is potentially the major affecting our analysis, but recent studies suggest it is difficult to determine how it may impact genotype-phenotype associations, particularly in meta-analyses. While molecular/ physical traits are less likely to being affected by participation bias, recall in last date of menstrual period may have biased the estimates of gestational duration. Samples included in this study have been largely recruited in Nordic countries, where rates of preterm delivery are amongst the lowest in the world. |
| Ethics oversight | All participants provided a signed written informed consent and study protocols were approved by each respective Institutional review boards. Ethic statements from each participating cohort are detailed in the Supplementary Note. Briefly, approval of study protocols was obtained from Ethical and Independent Review Services (https://eandireview.com/), Ethics Committee and the Local Research Ethics Committees (Bristol and Weston, Southmead, and Frenchay Health Authorities), Bradford Local NHS Research Ethics Committee, South East Multi-centre Research Ethics Committee and the Joint UCL/UCLH Committees on the Ethics of Human Research (Committee A), Institutional Review Board of the Children's Hospital of Philadelphia, the reproductive health protocol, the National Bioethics Committee (Iceland) following evaluation of the Icelandic Data Protection Authority, Regional Scientific Ethical Committee of the Region of Mid Jutland and the Danish Data Protection Agency, the Regional Scientific Ethical Committee of Copenhagen, the Ethics Review Committee of the University of Tartu, the North and East Devon (UK) Local Research Ethics Committee, the Ethics Committee of the Helsinki University |

Central Hospital, the Investigational Review Boards at all participating institutions of GPN, The Regional Committee for Medical Research Ethics, the Regional Committee for Medical Research Ethics, Southern Norway, Oslo, Norway, the CHUS ethic committee board, the Regional Ethics Committee for Medical Research Ethics South East Norway and the Women's and Newborn Health Service.

Note that full information on the approval of the study protocol must also be provided in the manuscript.

# Field-specific reporting

Please select the one below that is the best fit for your research. If you are not sure, read the appropriate sections before making your selection.

☒ Life sciences          ☐ Behavioural & social sciences          ☐ Ecological, evolutionary & environmental sciences

For a reference copy of the document with all sections, see nature.com/documents/nr-reporting-summary-flat.pdf

# Life sciences study design

All studies must disclose on these points even when the disclosure is negative.

| | |
|---|---|
| Sample size | The research sample consisted of 195,555 samples for gestational duration, 276,218 samples for preterm delivery (18,797 cases) and 115,307 samples for preterm delivery (15,972 cases) from 18 different cohorts. We included previously published data from 23andMe (Zhang et al., 2017) as well as data from different cohorts from the Early Growth Genetics Consortium, the Estonian Biobank, the Danish Blood Donor Study Genomic Consortium, the Norwegian Mother, Father and Child cohort study, The Trøndelag Health Study and deCODE genetics (Iceland). To study the effects of the parental transmitted and non-transmitted alleles, we pooled together results from the largest parent-offspring datasets available to date (n = 136,833 parent-offspring trios or mother-child duos). These samples were partly included in the discovery analysis (main GWAS meta-analysis). Other data generated in this study are included in the main article or in the Supplementary Tables or Figures. <br><br> For analyses where Mendelian randomization was performed, we used to distinct, complementary methods: two- and one-sample Mendelian randomization. The former was chosen to use the largest sample size as possible, given that prior studies on the relationship between sex-hormones and gestational duration were lacking (limiting the calculation of study sample size). One-sample Mendelian randomization was mainly carried out by deriving polygenic scores from lead SNPs (genome-wide significant) for a number of traits (sex hormones and birth weight). This analysis was carried out primarily to detect whether the effect observed using two-sample Mendelian randomization was driven by the maternal or the fetal genome, and was mainly used as an exploratory analysis. With regards of the effects of fetal growth (fetal effects on birh weight) on gestational duration, previous evidence suggested an effect of 3 days per standard deviation in birth weight (n = 10,000). In our analysis, we pooled data in >32,000 parent-offsprings, which we consider was sufficient for a reduction in effect size of 50%. As a general note, power calculations for the effects of parental transmitted and non-transmitted alleles is complex for several reasons: previous evidence is difficult to gather, observed effects using phenotypic associations may be misleading (i.e., in cases of opposite effect direction between maternal and fetal genomes) and additive effects are assumed. |
| Data exclusions | Data exclusions were performed on a cohort-specific basis. Whenever the information was available, medically-initiated labors were excluded. We excluded pregnancies lasting <140 days (20 completed weeks) or >310 days (44 completed weeks), multiplets as well as women with health complications prior to or during pregnancy and congenital fetal malformations. |
| Replication | No direct replication were performed of the main results, given that we pooled data from most cohorts with gestational duration (or preterm delivery) and genotype data available. We replicated the previously reported gestational duration loci (Zhang et al., 2017) in an out-of-sample analysis. |
| Randomization | Not applicable - observational study. |
| Blinding | Not applicable - observational study. |

# Reporting for specific materials, systems and methods

We require information from authors about some types of materials, experimental systems and methods used in many studies. Here, indicate whether each material, system or method listed is relevant to your study. If you are not sure if a list item applies to your research, read the appropriate section before selecting a response.

## Materials & experimental systems

| n/a | Involved in the study |
|---|---|
| ☒ | ☐ Antibodies |
| ☒ | ☐ Eukaryotic cell lines |
| ☒ | ☐ Palaeontology and archaeology |
| ☒ | ☐ Animals and other organisms |
| ☒ | ☐ Clinical data |
| ☒ | ☐ Dual use research of concern |

## Methods

| n/a | Involved in the study |
|---|---|
| ☒ | ☐ ChIP-seq |
| ☒ | ☐ Flow cytometry |
| ☒ | ☐ MRI-based neuroimaging |

