## [Peer Review File · Nature Genetics]

Peer Review Information

Manuscript Title: Genetic effects on the timing of parturition and links to fetal birth weight

Corresponding author name(s): Dr Pol Sole-Navais, Prof Bo Jacobsson

Reviewer Comments & Decisions:

Decision Letter, initial version:
--

22nd June 2022

Dear Pol,

Your Article "Genetic effects on the timing of parturition and links to fetal birth weight" has been seen by three referees. You will see from their comments below that, while they find your work of interest, they have raised several relevant points. We are interested in the possibility of publishing your study in Nature Genetics, but we would like to consider your response to these points in the form of a revised manuscript before we make a final decision on publication.

To guide the scope of the revisions, the editors discuss the referee reports in detail within the team, including with the chief editor, with a view to identifying key priorities that should be addressed in revision and sometimes overruling referee requests that are deemed beyond the scope of the current study. In this case, we ask that you address all technical queries related to the association analyses and their interpretation, extend other analyses (e.g. eQTL colocalization, gene prioritization, polygenic scores, evidence for co-adaptation, etc.) where feasible, and revise the presentation for clarity throughout. We hope you will find this prioritized set of referee points to be useful when revising your study. Please do not hesitate to get in touch if you would like to discuss these issues further.

We therefore invite you to revise your manuscript taking into account all reviewer and editor comments. Please highlight all changes in the manuscript text file. At this stage we will need you to upload a copy of the manuscript in MS Word .docx or similar editable format.

*2) If you have not done so already please begin to revise your manuscript so that it conforms to our Article format instructions, available [here](http://www.nature.com/ng/authors/article_types/index.html). Refer also to any guidelines provided in this letter.

[redacted]

We hope to receive your revised manuscript within 8-12 weeks. If you cannot send it within this time, please let us know.

Sincerely,
Kyle

Kyle Vogan, PhD
Senior Editor
Nature Genetics
<https://orcid.org/0000-0001-9565-9665>

Referee expertise:

Referee #1: Genetics, reproductive traits, statistical methods

Referee #2: Genetics, reproductive traits, metabolic traits

Referee #3: Genetics, fetal growth, metabolic traits

Reviewers' Comments:

Reviewer #1:
Remarks to the Author:

This is a comprehensive and ambitious study, which is potentially important, but for the same reason, I have had hard time to understand what the most important findings are, how strong the evidence is, and how appropriate the approaches are. The abstract is so descriptive that it did not alleviate my concerns.

The title abstract and the discussion section suggest that the authors were most interested in the genetics of the timing of parturition. However, the first paragraph of the introduction ends with "To date, studies of genetic variation have not reported distinct effects on different phenotypes of the timing of parturition." First, it is difficult to ascertain whether it is an accurate statement. Second, it seems to suggest that distinct genetic effects on different phenotypes of the timing of parturition may be the primary objective of this work, which is re-enforced by the several questions posed in the last paragraph of the introduction. Furthermore, many complex steps of analysis also seem to be aimed at answering those questions. Although those questions are important, and methods exist and are reasonable, I did not find those questions were convincingly addressed.

The following are my specific questions to illustrate why the answers weren't clear or convincing to me.

The largest sample size is a distinct advantage of this work, as noted by the authors that it is "a four-fold increase in sample size compared to the largest published maternal GWAS of gestational duration." However, the sample size is one, albeit important, aspect. The potential heterogeneity in the large number of cohorts also deserves attention in terms of its impact on the findings.

The authors included a total of 18 different cohorts. It would be helpful to have a supplemental table that lists the cohort, the number of samples used, and the major citation. I might have missed it, but I did look for it. This is important and useful because the cohort selection is the foundation of the findings. The authors already cited 66 references. In my opinion, if the authors used a major cohort, they must at least cite a major reference, instead of leaving it in a supplemental text. I noted that

some groups are included in the authorship such as Estonian Biobank Research Team, Danish Blood Donor Study Genomic Consortium, and Early Growth Genetics Consortium, but most groups were not included, acknowledged, or cited in the main text. What was the rationale?

In the "Meta-analysis of genome-wide association studies," the authors said that "After meta-analysis, we removed genetic variants not available for at least half the maximum sample size, resulting in 9-10 million genetic variants." What does the maximum sample size refer to? Why half?

Also, it was noted "LD-score regression intercepts were substantially lower than genomic inflation factors, suggesting that the inflation in test statistics was mostly due to polygenicity. Test statistics were not further adjusted for genomic control for any of the phenotypes." Why was it mostly due to polygenicity, but not the heterogeneity of the cohorts? Did the authors consider the inclusion of some leading principal components of the genomic markers? If so, did they make any difference in the conclusions, and if not, why?

In the "Colocalization" analysis, a Bayesian approach was used, and the inference was based on the posterior probability of five hypotheses. I can follow the probability of colocalization, but can't see the jump from the "Association with both phenotypes" to "the causal variant is shared."

In the effort of "Resolving effect origin," a regression model was used by considering the maternal non-transmitted and transmitted alleles. How were the non-transmitted and transmitted alleles inferred or separated, especially when there were ambiguities? What were the "covariates"? Was more than one model considered and did the conclusions vary?

For the "Locus pleiotropy at 3q21" analysis, FINNGEN and Pan UK Biobank data were used? Why were other cohorts excluded?

In the "Latent causal variable analysis," the authors "defined $GCP \geq 0.6$ between two traits as evidence of full or nearly full genetic causality." Is this a good practice to declare the full genetic causality? What do we gain other than stating the direct evidence from the data?

In the "Gestational duration polygenic score analysis," the MoBa cohort played a distinct role. Why was it unique from the other cohorts? Receiver operating characteristic, and area under the curve were calculated for the diagnosis of prediction accuracy of preterm delivery. What is the final prediction model, and how accurate is it?

In the "Multi-trait conditional analysis," the authors obtained birth weight summary statistics from four different GWAS within the EGG Consortium. Why were more cohorts not used?

Most references suggest that genetic correlation varies between -1 and 1. The authors reported a perfect genetic correlation of 1.17. If the upper limit is allowed to exceed 1, how do we know that 1.17 is perfect?

Reviewer #2:
Remarks to the Author:

This is a beautifully written piece of work that is interesting and appears methodologically rigorous and robust. The number of identified loci is very low compared to many complex traits, however I thought several of the analyses performed were highly insightful and a clear advance on other papers I have read in this area. I have a few minor comments for consideration:

1. Two loci were identified via approximate conditional analysis implemented in GCTA. From experience I've found this approach can often identify fairly spurious associations if the signals share LD with other signals, or if the beta estimates change considerably between pre and post conditional analysis. It's not clear if these issues have already been considered, but if not the authors may wish to explore this further.
2. Did all of the previously reported loci associated with gestational duration replicate?
3. I couldn't quite see the logic in only using iPSCs for eQTL mapping after reading the reason given in the methods. Did the authors explore whether GTEx tissues were enriched using LDSC? It might prove valuable to incorporate additional tissues into the eQTL colocalization work.
4. I thought slightly more work could be put in to mapping putatively causal genes to variants, such as using protein-QTL resources and fine-mapping coding variants.
5. Can the authors use approaches such as genomic SEM or MTAG to leverage the high genetic correlation between gestational duration / pre-term birth / maternal influences on BW, to identify additional novel loci?
6. The authors may wish to consider using previously defined clusters of sex-hormone acting variants to better discriminate between the effects of testosterone and SHBG (see Ruth et al paper already cited).
7. The fact that age at menopause appears in the LCT analysis seems to undermine the approach somewhat – presumably the MR of this is not associated?
8. I found some of the text in the section "Genetic association between gestational duration and birth weight" a little hard to follow and it might be worthwhile rewriting some bits for added clarity. For example, I was a little confused by the sentence "The genes tagging the SNPs with a modest reduction in effect size after conditioning (relative difference of effect > -0.2) were enriched in GO terms related to glucose and insulin metabolism and KEGG pathways also related to insulin and type II diabetes". I would expect that maternally acting BW genes that have effects mediated by gestational duration are NOT associated with insulin metabolism – is that the case? If not, how do the authors speculate insulin action impacts gestational duration?
9. Not all results that appear in supplementary tables have been mentioned in the main text (e.g. dominant and recessive models).
10. Some supplementary tables might be misnumbered (e.g I think S14 should be S13 in the text).

Reviewer #3:

Remarks to the Author:

Solé-Navais et al investigated the genetic basis of gestational duration, preterm birth, and post-term birth in a GWAS meta-analysis framework by assembling several birth cohorts of European ancestry individuals. The authors address a major data gap in the human gestational length physiology and in the pre-term birth problem that is a well-known, yet largely unaddressed, pregnancy complication leading to prenatal and childhood mortality and co-morbidities. This is by far the largest GWAS of gestational duration. The study identified 22 associated genetic loci for gestational duration, 6 for pre-term birth, and 1 for post-term birth; several loci were novel, despite their small effects (maximum effect on gestational duration was 27 hours; maximum odds of pre-term birth was 1.1). An important additional contribution of the study was the attempt to differentiate between maternal and fetal origin of the effect of the loci – this analysis clustered several loci with modest to high confidence. Further analysis was done to characterize genetic links between parturition and birth weight; this has shed some useful insights. In summary, this is an important advance to the genetic architecture of gestational duration in individuals from European populations, but unfortunately, does not address the missing genetic data on non-European populations that bear a higher burden of pre-term birth and its sequelae.

1. Page 10-11. PRS developed for gestational duration was tested for association with pre-term birth. Apparently, this contradicts the findings described earlier: weak genetic correlation was found between gestational duration and pre-term birth, and only few overlapping loci were identified. The authors suggested that the genetic architecture of gestational duration is likely to be different from that of pre-term birth. Why was then PRS developed and validated for gestational duration used to predict pre-term birth, and how valid is the modest prediction as found in an AUC of 0.61? Shouldn't a separate PRS be developed and tested for pre-term birth?
2. One of the main findings presented in the abstract as well as other parts of the paper is the birth weight-lowering fetal effect for maternal alleles associated with longer gestational duration. This finding is presented in Fig 4C & Table S13. a) Although the regression line shows significant inverse correlation, fetal-only effects are positive for 41% (9/22) of the SNPs evaluated. Compared to other data presented in the paper (e.g. Fig S11, presenting more consistent effects across most loci for maternal effects on gestational duration and birth weight), this evidence is less strong. Can additional supporting evidence be presented? b) As suggested by the authors, this findings may be in line with the co-adaptation theory. A stronger case can be made with further examination of birth weight in mom-baby pairs in which gestation-increasing maternal alleles are present in both the mother and fetus, vs pairs in which those alleles are present in the mom but not in the fetus. c) Do the loci linked to pre-term delivery suggest a tip-off of co-adaptation leading to "conflict" or failure-to-adapt? This may shed a more clinically relevant insight.
3. A major limitation of the study is its exclusive focus on European ancestry individuals in a clinical condition (pre-term birth) more prevalent in some populations of non-European ancestry, but sadly this limitation has not even been acknowledged.
4. The section presenting an attempt to highlight evolutionary dynamics is interesting, but needs more clarification. What was the rationale behind limiting the evolutionary analysis to the gestational duration loci colocalized with iPSC eQTLs? Do the findings offer hint at how evolutionary selection favored longer gestational duration in humans (e.g. TET3 and ZBTB38 index SNPs' ancestral alleles with high population frequency and conservation scores associated with longer gestational duration)?

What is the rationale behind the Fst-based population differentiation test in the context of pre-term birth which is prevalent globally? What does the significant Fst difference between African and East Asian population mean based on the relatively high prevalence of pre-term birth globally? ...and given the unknown transferability of this European ancestry-based association to those two population groups?

5. Why was colocalization done in iPSC? Colocalization in GTEx tissues may be useful given the apparently relevant female reproductive tissue.

6. Based on RNA-based enrichment of reproductive tissues, the authors said that the genetic effects may be timed towards end of gestation (page 8). I find it difficult to understand how this claim is supported by the presented data. Can one exclude the possibility that the loci can have effects on early pregnancy physiological changes to the reproductive tissues, but is less strong to induce miscarriage? A re-visit to this statement or further supportive evidence is warranted.

7. Page 14-15: With COJO analysis, a gestational duration-mediated effect of maternal genetic loci on birth weight has been found using 87 maternal genome SNPs previously found to have suggestive association with birth weight. This is an interesting finding. Can the authors validate the finding showing that maternal effects on birth weight are partially due to effects on gestational-duration using the sets of ~30 variants with stronger evidence for belonging in the "maternal only effect" cluster by Warrington as well as the more recent study by the deCODE group?

Minor comments

1. On page 19, the second paragraph cites Table S14, which should have been Table S13.
2. In the abstract, the sentence on "...sex-specific..." is not clear unless a reader refers to the Results section. Please rephrase this with a more direct presentation of the finding.
3. On page 8 (results), for ease of reading, please add the gestational duration for controls.
4. Page 14: Clarify what is meant by 'limited adjustment for gestational duration', whether this context should be considered in the interpretation of the findings/conclusions.
5. Figure S3: Add a legend describing the colors of the text (tissues) and dots
6. Figure S9: Edit the 6th line of the legend
7. Figure S13: Label the plots with names of the three gene regions
8. The first paragraph of the introduction ends with a note that sets an expectation that the current work dissected differential genetic effects on clinical subtypes of parturition timing. My expectation was to see genetic links with subtypes such as PROM, spontaneous onset labor etc, but no such findings were presented. This section and the discussion of this topic in the Discussions section needs tempering or findings of clinical subtypes, if any, added.

Author Rebuttal to Initial comments**Responses to the Editor and Reviewers**

To guide the scope of the revisions, the editors discuss the referee reports in detail within the team, including with the chief editor, with a view to identifying key priorities that should be addressed in revision and sometimes overruling referee requests that are deemed beyond the scope of the current study. In this case, we ask that you **address all technical queries related to the association analyses and their interpretation, extend other analyses (e.g. eQTL colocalization, gene prioritization, polygenic scores, evidence for co-adaptation, etc.) where feasible, and revise the presentation for clarity throughout.** We hope you will find this prioritized set of referee points to be useful when revising your study. Please do not hesitate to get in touch if you would like to discuss these issues further.

We thank the editors and reviewers for the positive assessment of our work, and highly appreciate the comments and suggestions. We have used every opportunity to revise the manuscript, and consider it greatly improved. Specifically, we have addressed all technical queries, extended gene prioritization strategies (including eQTL and protein-QTL colocalization and genome-wide enrichment analyses) and built a polygenic score for preterm delivery. While revising the presentation throughout, we have rearranged several sections. To note, we have merged the section describing the effects of a fetal growth polygenic score on gestational duration and the antagonistic pleiotropy analysis (i.e., suggestion of co-adaptation). In this same section, we also provide evidence that the antagonistic effects are limited to the maternal gestational duration increasing alleles.

Reviewers' Comments

Reviewer #1:

Remarks to the Author:

This is a comprehensive and ambitious study, which is potentially important, but for the same reason, I have had hard time to understand what the most important findings are, how strong the evidence is, and how appropriate the approaches are. The abstract is so descriptive that it did not alleviate my concerns.

The title, abstract and the discussion section suggest that the authors were most interested in the genetics of the timing of parturition. However, the first paragraph of the introduction ends with “To date, studies of genetic variation have not reported distinct effects on different phenotypes of the timing of parturition.” First, it is difficult to ascertain whether it is an accurate statement. Second, it seems to suggest that distinct genetic effects on different phenotypes of the timing of parturition may be the primary objective of this work, which is re-enforced by the several questions posed in the last paragraph of the introduction. Furthermore, many complex steps of analysis also seem to be aimed at answering those questions. Although those questions are important, and methods exist and are reasonable, I did not find those questions were convincingly addressed.

We thank the reviewer for the critical revision of our work, and for the constructive comments and suggestions. These have triggered many interesting discussions, and an even more careful inspection of several technical aspects. We have streamlined the abstract and introduction to provide a clear message, removed confusing statements, and, overall, revised the text for readability.

The following are my specific questions to illustrate why the answers weren't clear or convincing to me.

1. The largest sample size is a distinct advantage of this work, as noted by the authors that it is “a four-fold increase in sample size compared to the largest published maternal GWAS of gestational duration.” However, the sample size is one, albeit important, aspect. The potential heterogeneity in the large number of cohorts also deserves attention in terms of its impact on the findings.

Heterogeneity of cohorts is an important point. For instance, the use of different methods for estimating gestational duration may limit our ability to identify certain effects. We have now mentioned this in the Discussion. Heterogeneity in cohorts may translate into heterogeneity in the effect sizes estimated.

We wanted to examine whether heterogeneity, estimated using the I^2 statistic, was widespread for our index SNPs. In the previous version, we tested for heterogeneity using Cochran's Q-test (detailed in methods), but we failed in providing these results. We have now added I^2 and p-value for heterogeneity for all index variants in Supplementary Table 2. Of the 24 gestational duration index SNPs, only five show evidence of heterogeneity; all of them were previously reported to be associated with gestational duration (*WNT4*, *EBF1*, *ADCY5*, *EEFSEC* and *AGTR2*). As exemplified below, heterogeneity is likely to be

higher in these loci due to winner's curse, rather than overall heterogeneity between cohorts (Supplementary Fig. 2 and below the example of *EBF1*).

A visual inspection of these signals using forest plots shows that heterogeneity is largely driven by one of the major cohorts. Here, we show the example of the lead SNP at the *EBF1* gene region (SNP with the lowest association p-value with gestational duration and the largest heterogeneity statistic, $I^2 = 0.67$). Using random-effects meta-analysis, the effect estimate remains almost unchanged (beta = -0.83; 95%CI = -1.08, -0.58) and the p-value is still genome-wide significant (p-value = 3.1×10^{-10}). As can be observed in the plot, most of the heterogeneity is driven by 23andMe data, where this signal was previously identified. Removing 23andMe data from the meta-analysis reduces heterogeneity from 0.67 to 0.16 (p-value = 0.118), and the association with gestational duration remains similar to that presented in the manuscript (excluding 23andMe: beta = -0.69; 95%CI = -0.84, -0.53; p-value = 3.1×10^{-18} ; including 23andMe: beta = -0.77; 95% CI = -0.87, -0.67; p-value = 2.4×10^{-47}).

Forest plot for locus Chr 5: EBF1

Lead variant: 5:157895049:C:T:SNP

Meta-analysis: Beta= -0.77 (95% CI= -0.874, -0.665); pvalue= 2.44e-47

Test for heterogeneity: $I^2 = 67.1\%$; Het pvalue= 0.0002671

Figure 1. Forest plot of the leading SNP on gestational duration at *EBF1* locus. Each square represents the effect size for a particular cohort and error bars are the 95% CI. Diamond represents the estimate after meta-analysis.

2. The authors included a total of 18 different cohorts. It would be helpful to have a supplemental table that lists the cohort, the number of samples used, and the major citation. I might have missed it, but I did look for it. This is important and useful because the cohort selection is the foundation of the findings. The authors already cited 66 references. In my opinion, if the authors used a major cohort, they must at least cite a major reference, instead of leaving it in a supplemental text. I noted that some groups are included in the authorship such as Estonian Biobank Research Team, Danish Blood Donor Study Genomic Consortium, and Early Growth Genetics Consortium, but most groups were not included, acknowledged, or cited in the main text. What was the rationale?

Cohorts were previously listed in Supplementary Table 14. Based on this comment and to increase its visibility, the table is now the first supplementary table mentioned in results (Supplementary Table 1). In the Methods section, under the “Study cohorts and individual-level GWAS” heading, we listed the major cohorts, but we did not provide a reference for each. We thank the reviewer for noticing this; in the revised manuscript we provide such reference.

With regards to acknowledgements, we believe all participants deserve the same recognition, irrespective of the sample size of the study they participate in. For this reason, we will keep acknowledgements for all cohorts in the Supplementary Text.

3. In the “Meta-analysis of genome-wide association studies,” the authors said that “After meta-analysis, we removed genetic variants not available for at least half the maximum sample size, resulting in 9-10 million genetic variants.” What does the maximum sample size refer to? Why half?

We agree with the reviewer that “the maximum sample size” requires additional context. We have revised this sentence, and now reads as:

“After meta-analysis, we removed genetic variants reported in less than half the number of available samples for each phenotype, resulting in 9-10 million genetic variants.”

The maximum sample size (after meta-analysis) referred to the SNP available in the largest number of samples, for each phenotype (gestational duration, preterm and post-term). To homogenize the summary statistics provided by the different groups, we decided to keep, after meta-analysis, genetic variants that were available in at least half the total sample size for each phenotype. For instance, for gestational duration, the variant with the largest sample size was available in 195,555 women; before QC, variants with as “little” as 300 samples were present, which could lead to cohort-specific effects, and large differences in power. We chose “half” because it enforces the presence of the variant summary metrics in at least two cohorts for each of the phenotypes.

4. Also, it was noted “LD-score regression intercepts were substantially lower than genomic inflation factors, suggesting that the inflation in test statistics was mostly due to polygenicity. Test statistics were not further adjusted for genomic control for any of the phenotypes.” Why was it mostly due to polygenicity, but not the heterogeneity of the cohorts? Did the authors consider the inclusion of some leading principal components of the genomic markers? If so, did they make any difference in the conclusions, and if not, why?

Analysis was performed individually by each cohort, but with a centralized protocol, which stressed the use of principal components or mixed models to deal with population stratification. Our view is that genomic inflation factor was small for all three phenotypes (<1.1) and the LD-score regression intercept was even lower, which is consistent with pleiotropy (Bulik-Sullivan, *Nature Genetics*, 2015). As mentioned in a previous comment (see #1), heterogeneity was limited to the loci that were previously reported (5 of 24), so we do not consider it to be pervasive at the genome-wide scale.

5. In the “Colocalization” analysis, a Bayesian approach was used, and the inference was based on the posterior probability of five hypotheses. I can follow the probability of colocalization, but can’t see the jump from the “Association with both phenotypes” to “the causal variant is shared.”

The colocalization analysis method that we have used tests for the five hypotheses described in “Methods” at the locus level. At loci where there is at least one significant association, LD will generally draw a “pattern” of association around the top SNP (not necessarily the causal one). Colocalization analysis tests whether the patterns of association for two traits, at a single locus, match one another. Particularly, hypothesis 4 describes an association for the two phenotypes (at the same locus), but the causal variant is not shared (i.e., the associations are driven by distinct SNPs with low LD, patterns of association mismatch). Hypothesis 5 describes the case where there is an association for the two

phenotypes and the pattern of association matches the two. While we do not know which of the SNPs is the causal one (maybe it is not even part of the summary statistics), matching patterns of association would suggest the sharing of the same causal variant, assuming there is only one causal variant in the locus.

The two hypotheses (sharing locus with distinct causal variant, sharing locus and causal variant) are illustrated in this figure from a recent colocalization publication (Zuber V, AJHG, 2022):

Figure 2. Schematic representation of colocalization showing two scenarios. In both scenarios there is an association on the two phenotypes at the same locus. Panel A shows the scenario when the association is driven by variants in low LD, panel B and C show the scenario when the associations are driven by variants in close LD.

In the first panel, the locus is shared between the two loci, but the causal variant is not shared. The second panel shows sharing of the same locus, and sharing of the causal variant (despite not knowing which is the causal variant).

We have now clarified this in the “*Online Methods*” section and have refrained from using the term “causal”.

6. In the effort of “Resolving effect origin,” a regression model was used by considering the maternal non-transmitted and transmitted alleles. How were the non-transmitted and transmitted alleles inferred or

separated, especially when there were ambiguities? What were the “*covariates*”? Was more than one model considered and did the conclusions vary?

Transmitted and non-transmitted alleles were determined by statistically phasing genotype data. Generally, in this process, parental origin of each allele is inferred using genotypes of relatives, reference cohort data, or distributions of genotypes within the cohort and LD measurements. Specifically, in Icelandic data, parental alleles were inferred by combining long-range phasing and genealogy using the methods described in (Juliusdottir T, Nat Genet, 2021, Kong A, Nat Genet, 2008 and Kong A, Nature, 2009), which relies primarily on Mendelian rules to resolve the alleles; few ambiguities are present because many relatives are available for each proband. All other parent-offspring cohorts but HUNT were phased using SHAPEIT2 (O'Connell D, Plos Genetics, 2014), which initially infers haplotypes within the cohort using a hidden Markov model, and then uses relative information to correct errors and assign parents to haplotypes. Reliability of these methods is considered to be high, particularly when large amounts of identical-by-descent segments are present (i.e., when parent-offspring data is available). HUNT data was phased using Eagle v2.3 (Loh PR, Nat Genet, 2016), which is a combination of the two methods above mentioned (long-range phasing and hidden Markov model to iteratively refine phase calls). Overall, the amount of ambiguities is negligible, and whenever present, we should expect a decrease in statistical power.

We apologize for not having provided such information in the previous version: for clarity and transparency, we have now added this information in the Methods section, including a supplementary table (Supplementary Table 17) describing which phasing methods were used for each cohort, as well as the sample size.

Each cohort adjusted for the covariates that were considered necessary, which were: genotype principal components (from the fetus), parity, maternal age and batch. As can be noticed, the alleles tested were incorporated in the same model. However, running the linear regression models on each allele separately did not affect the estimates.

7. For the “Locus pleiotropy at 3q21” analysis, FINNGEN and Pan UK Biobank data were used? Why were other cohorts excluded?

For this analysis, we considered publicly available GWAS summary statistics from a broad spectrum of phenotypes and without a pre-specified hypothesis. To the best of our knowledge, UK Biobank and FINNGEN are the only large-scale biobanks conducting and publishing GWAS summary statistics in a systematic way. Also, a large part of the individuals in these two biobanks are more genetically similar to our samples than other biobanks (e.g., Biobank Japan), which is required for colocalization analyses. While we performed the analysis in only two cohorts, we covered >3,500 phenotypes, from ICD codes to biomarkers. We consider these to be sufficient, although we acknowledge some of these phenotypes may be of low quality, particularly those that are self-reported.

8. In the “Latent causal variable analysis,” the authors “defined $GCP \geq 0.6$ between two traits as evidence of full or nearly full genetic causality.” Is this a good practice to declare the full genetic causality? What do we gain other than stating the direct evidence from the data?

We thank the reviewer for this reflection. We have borrowed the term “full or nearly full genetic causality” from the researchers that developed the Latent Causal Variable model (see O’Connor L and Price AL, Nature Genetics, 2018) for a single reason: it simplifies the underlying message (i.e., “GCP” may not be known to the readers). We agree that one should be skeptical about making strong statements (e.g., “full causality”), and this is why we have consistently employed “full or nearly full” throughout the manuscript to describe genetic causality proportions ≥ 0.6 that were significantly different from 0. Not to say that setting hard thresholds is somewhat arbitrary. To frame the results and conclusions of that section we have set thresholds for both p-value and the genetic causality proportion estimate. In this sense, we have valued simplicity over precision. To complement the above-mentioned statement, we now provide the specific GCP values (or ranges).

9. In the “Gestational duration polygenic score analysis,” the MoBa cohort played a distinct role. Why was it unique from the other cohorts? Receiver operating characteristic, and area under the curve were calculated for the diagnosis of prediction accuracy of preterm delivery. What is the final prediction model, and how accurate is it?

To generate the polygenic score we required good-quality, individual-level data with a relatively large sample size. The leading junior/ senior authors have access to the MoBa cohort, among others, which is linked to the Medical Birth Registry of Norway from the years 2002-2008. Other large cohorts linked to Medical Birth Registries have data as early as the 70s, which comes at the expense of larger heterogeneity, and degrees of missingness in key information such as labor initiation, c-section, etc.

We apologize for not specifying the final prediction models that include the polygenic score; in the revised version we have included these for each polygenic score, which was the polygenic score itself. Models were tested using the first five principal components but none were significantly associated with gestational duration and therefore the simplest model was deemed the most appropriate. We used the following two regression models for the polygenic score of gestational duration and the second model for the polygenic score of preterm delivery:

Gestational Duration ~polygenic score (Linear Model)

Preterm delivery ~polygenic score (Logistic Model)

10. In the “Multi-trait conditional analysis,” the authors obtained birth weight summary statistics from four different GWAS within the EGG Consortium. Why were more cohorts not used?

An increase in sample size in the birth weight GWAS would have been highly valuable, however, in this analysis we were interested in the relationship between gestational duration and birth weight. This forced us to limit the analysis to the largest GWAS of birth weight that was largely not adjusting for gestational duration (>80% of the sample size). To note, while this GWAS of birth weight was performed within the EGG Consortium, it included data from mothers, such as the UK Biobank. We are aware of a recent manuscript (Juliusdottir T, Nat Genet, 2021) that has a larger sample size than the one we have employed. However, Juliusdottir et al meta-analysed the data we have used and their own GWAS data, which was adjusted for gestational duration. We have now clarified the rationale both in the “Results” and “Online Methods” section.

In the revised manuscript, we have performed the same analyses using individual level data from Iceland and Norway, using SNP dosage and the parental transmitted and non-transmitted alleles, respectively. We provide these results, which support the findings obtained using summary statistics, in Supplementary Fig. 14 and Supplementary Table 13

11. Most references suggest that genetic correlation varies between -1 and 1. The authors reported a perfect genetic correlation of 1.17. If the upper limit is allowed to exceed 1, how do we know that 1.17 is perfect?

Theoretically, the genetic correlation should be bound to $[-1, 1]$, as it has the same meaning as the common statistical correlation, applied to genetic effects. We agree with the reviewer that most references report genetic correlations between -1 and 1. This is exemplified in the original LD Score regression manuscript describing genetic correlations (Bulik-Sullivan, 2015, *Nature Genetics*), where figure legends are bound to $[-1, 1]$.

However, the estimation method in LD Score regression does not bound genetic correlations, and for very high correlations can thus produce estimates outside of $[-1, 1]$ due to sampling variation. (Note for example that the Supplementary Table 4 of the above reference shows 21 estimates of genetic correlation above 1). The genetic correlation was very high, with the lower bound of the 95%CI at 0.93.

Reviewer #2:

Remarks to the Author:

This is a beautifully written piece of work that is interesting and appears methodologically rigorous and robust. The number of identified loci is very low compared to many complex traits, however I thought several of the analyses performed were highly insightful and a clear advance on other papers I have read in this area. I have a few minor comments for consideration:

We thank the reviewer for the positive assessment of our work, and for providing useful comments. We have implemented most suggestions, which we think have greatly improved the manuscript.

1. Two loci were identified via approximate conditional analysis implemented in GCTA. From experience I've found this approach can often identify fairly spurious associations if the signals share LD with other signals, or if the beta estimates change considerably between pre and post conditional analysis. It's not clear if these issues have already been considered, but if not the authors may wish to explore this further.

We consider this to be a relevant issue. The conditionally independent signal at the *EBF1* gene region was previously reported, but was identified using the same method (although a different LD reference panel, Zhang G, NEJM, 2017).

To assess whether our results hold, we estimate the effect size on gestational duration of the secondary SNP with and without including the index SNP as a covariate in a linear regression model MoBa samples ($n = 19,493$ mothers) for *EBF1* and *KCNAB1* loci. We then compared the fold change in effect size to the one presented in the manuscript (i.e., using COJO in the full meta-analysis, Supplementary Fig. 2). As can be seen in Table 1 below, the fold changes with / without conditioning are very similar using individual level data and COJO analysis (0.67 vs 0.55 at *EBF1* and 0.01 and 0.01 at *KCNAB1*, respectively). Owing to this, we do not consider the conditionally-independent signals to be false-positives. In addition, while the fold change is negligible for the *KCNAB1* locus, the lead SNP (rs4359773) and the conditionally-independent SNP are not in LD ($R^2 = 0.0001$) and the latter was genome-wide significant before conditioning.

Table 1. Effect size on gestational duration of the secondary SNPs with and without conditioning on the index variant at the EBF1 and KCNAB1 loci using either individual level data (MoBa samples) or the meta-analysis summary statistics (COJO).					
Locus	Data	Conditioning	Beta	SE	Fold change
EBF1	Individual level	No	-0.19	0.10	0.67
		Yes	-0.33	0.11	
	Summary statistics	No	-0.23	0.05	0.55
		Yes	-0.36	0.05	
KCNAB1	Individual level	No	-0.30	0.11	0.01
		Yes	-0.30	0.11	

	Summary statistics	No	-0.27	0.05	0.01
		Yes	-0.27	0.05	
For the analysis using individual level data, we estimated the effect of the secondary signal on gestational duration using linear regression with or without including the index SNP as a covariate in MoBa samples (n = 19,493 mothers). For the COJO analysis, we used summary statistics from the meta-analysis presented in this work (n = 195,555 mothers).					

2. Did all of the previously reported loci associated with gestational duration replicate?

In the initial submission, we only reported that the *EEFSEC* locus replicated at nominally significant level after excluding 23andMe data, but that was regarding the lead SNP in the current meta-analysis. In an out-of-sample analysis (i.e., excluding 23andMe data), all four previously reported SNPs (the two in chromosome X were not present in our data) replicate at nominal significance. At the locus level (± 250 kb from previous lead SNP), all loci have a significant signal (including the ones on chromosome X, p-value $< 1e-5$). These results are shown in Supplementary Table 3.

3. I couldn't quite see the logic in only using iPSCs for eQTL mapping after reading the reason given in the methods. Did the authors explore whether GTEx tissues were enriched using LDSC? It might prove valuable to incorporate additional tissues into the eQTL colocalization work.

The decision to use iPSCs was based on the higher power that these eQTLs have compared to GTEx female reproductive tissues and on its resemblance to such tissues. However, based on your comment and that of Reviewer #3, we have updated our analyses to include both a tissue-specific RNA heritability enrichment (tissue-agnostic) using stratified LD-score regression and colocalization with eQTLs from vagina, uterus and ovary (GTEx) and endometrium (Mortlock S, Hum Reprod, 2020).

The tissue-specific enrichment we observed for the lead variants is supported by an enrichment in the same/ similar GTEx tissues (although only at nominal significance), expanding the evidence to the genome-wide scale. These results are now reported in Supplementary Fig. 4.

Regarding GTEx colocalization analyses, we identify one additional signal that colocalizes with protein coding gene eQTLs in the uterus (*ADCY5*). Overall, we see that the number of colocalizing loci increases with increasing sample size of the eQTL analysis, as could be expected. We have incorporated these results in Supplementary Table 4, and have updated the text accordingly.

4. I thought slightly more work could be put in to mapping putatively causal genes to variants, such as using protein-QTL resources and fine-mapping coding variants.

We want to thank the reviewer for this and the previous comment; their implementation has substantially improved the prioritization of candidate genes.

Initially, we looked-up for missense variants that were in moderate LD ($R^2 \geq 0.6$) with the lead SNPs identified, but unfortunately no such variants exist. This information was not described in the previous version of the manuscript and is now provided in the revised version.

As mentioned in the answer to the previous comment, we have now expanded the colocalization analysis with eQTLs to other tissues, which has improved the prioritization of genes. Furthermore, we have looked-up whether the gestational duration lead SNPs were protein QTLs in one of the the largest, publicly-available blood pQTL study (Pietzner*, Wheeler* et al., Science, 2021). Whenever one such lead SNP was a pQTL (suggestive evidence, $p\text{-value} < 5e^{-6}$), we performed a colocalization analysis with our GWAS on gestational duration. This information is now included in Supplementary Table 4. We want to highlight the colocalization between gestational duration and eQTLs for *OPRL1* and protein QTLs for POMC. Both *OPRL1* and *POMC* are related to pain perception, and *OPRL1* has been shown to control the inhibition of uterine contractions.

5. Can the authors use approaches such as genomic SEM or MTAG to leverage the high genetic correlation between gestational duration / pre-term birth / maternal influences on BW, to identify additional novel loci?

This is an attractive idea and we have discussed its feasibility in the context of this manuscript. Given the complex interplay between the maternal and fetal genomes on gestational duration and birth weight, we

must admit that answering this question would require the development of a specific analytical approach. We fear that this is beyond the scope of our current study.

6. The authors may wish to consider using previously defined clusters of sex-hormone acting variants to better discriminate between the effects of testosterone and SHBG (see Ruth et al paper already cited).

We have further explored the effect of testosterone, independent of SHBG, on gestational duration and preterm delivery. Nonetheless, we want to highlight that, in the previous version, we mentioned that the genetic correlations observed were likely due to pleiotropy (*“Testosterone and SHBG levels have a complex genetic link with the timing of parturition, likely explained by partial causality, as pointed out by the LCV analysis on gestational duration.”*). In the revised version of the manuscript, we show that the significant association observed using Mendelian randomization analysis may arise due to a violation of its assumptions (i.e., linkage disequilibrium between causal variants or horizontal pleiotropy, Extended Data Fig. 4 and Supplementary Fig. 12).

7. The fact that age at menopause appears in the LCT analysis seems to undermine the approach somewhat – presumably the MR of this is not associated?

We agree with the reviewer that adding age at menopause in the LCV analysis was misleading. Given that this analysis was data-driven, we only included age at menopause because it was genetically correlated with preterm delivery. We expected no causal associations between age at menopause and gestational duration, and therefore we did not perform the MR analysis for the effects of age at menopause on gestational duration/ preterm delivery. In the revised version, we have limited the LCV analysis to sex-hormones, which we have followed-up with a two-sample MR and polygenic scores using transmitted and non-transmitted alleles. Consequently, we have removed panel B from Fig. 3 - LCV results are reported in Supplementary Table 8.

8. I found some of the text in the section “Genetic association between gestational duration and birth weight” a little hard to follow and it might be worthwhile rewriting some bits for added clarity. For example, I was a little confused by the sentence “The genes tagging the SNPs with a modest reduction in effect size after conditioning (relative difference of effect > -0.2) were enriched in GO terms related to glucose and insulin metabolism and KEGG pathways also related to insulin and type II diabetes”. I would expect that maternally acting BW genes that have effects mediated by gestational duration are NOT

associated with insulin metabolism – is that the case? If not, how do the authors speculate insulin action impacts gestational duration?

We apologize for the confusion in this section. We have now simplified the text, and included further evidence by performing the analysis using both summary statistics (same method as previously, multi-trait COJO) and individual level data (using genotype dosage and parental transmitted and non-transmitted alleles) from Norway and Iceland on top variants classified as having a “Maternal Only” or “Fetal Only” effect (Warrington N, Nat Genet, 2019). This information is now part of Supplementary Fig. 14 and Supplementary Table 13.

In the enrichment analysis, we tried to understand the reasons behind an increase in the SNP effect size on birth weight after adjusting for gestational duration (a handful of SNPs). Studies have shown that elevated maternal blood glucose could increase birth weight but reduce gestational duration (see Chen J, Plos Medicine, 2020). Therefore, the maternal alleles associated with increased birth weight through an effect on maternal plasma glucose could have a negative effect on gestational duration and hence indirectly reduce birth weight. Thus, accounting for gestational duration in the maternal GWAS of birth weight will increase the effect size estimates of these variants.

As mentioned above, we replicated the reduction (on average) in effect size after conditioning on gestational duration for maternal variants and the null effect of conditioning on fetal variants. We also attempted to test the robustness of the enrichment analysis presented in this section, by varying the cut-off for selecting the SNPs with no or small decrease in effect size after conditioning (previously using “-0.2”). However, we were not able to replicate such results, and despite believing our above-mentioned arguments to be reasonable, we have removed the enrichment analysis from the results.

9. Not all results that appear in supplementary tables have been mentioned in the main text (e.g. dominant and recessive models).

We thank the reviewer for highlighting this, particularly in regards to the non-additive models. In an initial draft of the manuscript we had included such models, but after internal discussion within the working group, we decided to keep it outside of results. We failed to remove it from the supplementary file, but have done so in the revised manuscript. The reason for not including such analyses was mainly due to QC issues that made the non-additive models less reliable than the additive effect model. For example, for

dominant and recessive models, only the p-value was reported, effect/ reference alleles were only reported for the additive effect, the analysis was not carried out in all cohorts, etc.

In the revised version, we have made sure that all results in supplementary tables/ figures are also mentioned in the text.

10. Some supplementary tables might be misnumbered (e.g I think S14 should be S13 in the text).

We have now corrected this.

Reviewer #3:

Remarks to the Author:

Solé-Navais et al investigated the genetic basis of gestational duration, preterm birth, and post-term birth in a GWAS meta-analysis framework by assembling several birth cohorts of European ancestry individuals. The authors address a major data gap in the human gestational length physiology and in the pre-term birth problem that is a well-known, yet largely unaddressed, pregnancy complication leading to prenatal and childhood mortality and co-morbidities. This is by far the largest GWAS of gestational duration. The study identified 22 associated genetic loci for gestational duration, 6 for pre-term birth, and 1 for post-term birth; several loci were novel, despite their small effects (maximum effect on gestational duration was 27 hours; maximum odds of pre-term birth was 1.1). An important additional contribution of the study was the attempt to differentiate between maternal and fetal origin of the effect of the loci – this analysis clustered several loci with modest to high

confidence. Further analysis was done to characterize genetic links between parturition and birth weight; this has shed some useful insights. In summary, this is an important advance to the genetic architecture of gestational duration in individuals from European populations, but unfortunately, does not address the missing genetic data on non-European populations that bear a higher burden of pre-term birth and its sequelae.

We thank the reviewer for the careful inspection of our work, and for the positive assessment of it. We have taken every opportunity to improve the manuscript, and we acknowledge that we did not include a paragraph detailing limitations of our work, which is now provided in the Discussion.

1. Page 10-11. PRS developed for gestational duration was tested for association with pre-term birth. Apparently, this contradicts the findings described earlier: weak genetic correlation was found between gestational duration and pre-term birth, and only few overlapping loci were identified. The authors suggested that the genetic architecture of gestational duration is likely to be different from that of pre-term birth. Why was then PRS developed and validated for gestational duration used to predict pre-term birth, and how valid is the modest prediction as found in an AUC of 0.61? Shouldn't a separate PRS be developed and tested for pre-term birth?

We agree with the reviewer that the results obtained using genetic correlations were not phrased properly. We consider the genetic correlation between gestational duration and preterm delivery to be moderate-strong, and so is reflected in the large number of overlapping loci (all preterm delivery loci but one were identified in the GWAS of gestational duration). We have now rephrased the text to reflect that the genetic effects on gestational duration and preterm delivery are largely similar, and provide further evidence that the effect sizes of the index SNPs for the two phenotypes are highly correlated (Supplementary Fig. 10). In this sense, we considered only the PGS for gestational duration given its power, evidenced by the number of loci identified. The modest prediction accuracy of the gestational duration PGS is concordant with the R^2 on gestational duration.

However, we have discussed internally the use of a PGS of preterm delivery, and want to emphasize that this was indeed an interesting idea. We have thus added a PGS for preterm delivery, trained and validated in the same samples as the gestational duration PGS. Interestingly, we find that the preterm delivery PGS has a prediction performance on preterm delivery (AUC = 0.57; 95% CI = 0.51, 0.62) similar to that of the gestational duration PGS (AUC = 0.61; 95% CI = 0.55, 0.67), despite its lower power. These results are now included in Supplementary Table 6 and Supplementary Fig. 11.

2. One of the main findings presented in the abstract as well as other parts of the paper is the birth weight-lowering fetal effect for maternal alleles associated with longer gestational duration. This finding is presented in Fig 4C & Table S13. a) Although the regression line shows significant inverse correlation, fetal-only effects are positive for 41% (9/22) of the SNPs evaluated. Compared to other data presented in the paper (e.g. Fig S11, presenting more consistent effects across most loci for maternal effects on

gestational duration and birth weight), this evidence is less strong. Can additional supporting evidence be presented? b) As suggested by the authors, this findings may be in line with the co-adaptation theory. A stronger case can be made with further examination of birth weight in mom-baby pairs in which gestation-increasing maternal alleles are present in both the mother and fetus, vs pairs in which those alleles are present in the mom but not in the fetus. c) Do

the loci linked to pre-term delivery suggest a tip-off of co-adaptation leading to “conflict” or failure-to-adapt? This may shed a more clinically relevant insight.

In Fig. 4C we show that maternal alleles that are primarily associated with gestational duration tend to reduce birth weight through a fetal effect. We agree with the reviewer that the use of individual level data will be helpful to support such findings. However, in this precise analysis we have used GWAS summary statistics to precisely increase the power to identify such effects ($n = 136,000$ for gestational duration and 200K for own birth weight). To provide supporting evidence using individual level data, one could compare the effects of a gestational duration genetic score using the maternal transmitted and non-transmitted alleles on birth weight. We would expect the effects of the maternal transmitted genetic score (alleles of which are also present in the fetus, with a negative on birth weight) to be lower than that of the maternal non-transmitted genetic score (only present in the mother). From the results presented using summary statistics, we estimate this difference to be approximately 0.01 - 0.02 birth weight z-scores, or 5-10 grams, per day. In this manuscript, we have pooled together the largest sample size of parent-offsprings ($n > 40,000$), and still, this would not be enough to identify such small differences. Using an alpha level of 0.95, with a sample size of 45,000 parent-offsprings we would have a power of 24% to detect a 0.02 birth weight z-scores difference between the maternal transmitted and non-transmitted genetic scores.

With regards to point c), while these hypotheses are highly interesting, we recognize that the current differences in discovery between the preterm delivery and gestational duration GWAS are limited. In addition, the power of an analysis on preterm delivery using the transmitted/ non-transmitted alleles is much lower compared to that of gestational duration.

We have revised the text in results, methods and discussion sections to bring more clarity about the results, and highlight that the effects are due to pleiotropy (in opposite directions).

3. A major limitation of the study is its exclusive focus on European ancestry individuals in a clinical condition (pre-term birth) more prevalent in some populations of non-European ancestry, but sadly this limitation has not even been acknowledged.

We apologize for not acknowledging this limitation; we have now included a paragraph in the Discussion section describing this and other limitations. We are aware and are working on ongoing efforts to expand these analyses to diverse ancestries. We hope such analyses will provide many novel insights into unknown biology, and further our understanding of the genetic effects on the timing of parturition.

4. The section presenting an attempt to highlight evolutionary dynamics is interesting, but needs more clarification. What was the rationale behind limiting the evolutionary analysis to the gestational duration loci colocalized with iPSC eQTLs? Do the findings offer hint at how evolutionary selection favored longer gestational duration in humans (e.g. TET3 and ZBTB38 index SNPs' ancestral alleles with high population frequency and conservation scores associated with longer gestational duration)? What is the rationale behind the F_{st} -based population differentiation test in the context of pre-term birth which is prevalent globally? What does the significant F_{st} difference between African and East Asian population mean based on the relatively high prevalence of pre-term birth globally? ...and given the unknown transferability of this European ancestry-based association to those two population groups?

We limited the evolutionary analysis to the gestational duration loci that colocalized with eQTLs to try to identify similarities between these loci. In the revised manuscript, we have identified additional loci with strong evidence of colocalization between gestational duration and eQTLs. For this reason, we have now moved the section regarding evolutionary measures to the enrichment analysis section, and report the results for all loci (Supplementary Fig. 6) except the ones at the X chromosome and the HLA gene region). We highlight that there is not a single evolutionary force driving the associated loci, replicating previous findings.

5. Why was colocalization done in iPSC? Colocalization in GTEx tissues may be useful given the apparently relevant female reproductive tissue.

We want to thank the reviewer for this comment, which is in line with comments from Reviewer #2. We have addressed it in several ways, which we think have improved the prioritization of genes, among others. First, we have strengthened the evidence and expanded the enrichment from candidate loci to the genome-wide scale by applying stratified LD-score regression using tissue-specific gene expression (Supplementary Fig. 4). Second, we have performed colocalization analyses between gestational duration and eQTLs in additional tissues (Supplementary Table 4): ovary, vagina and uterus from GTEx and endometrium (Mortlock S, Hum Reprod, 2020). Third, we have performed colocalization with blood

protein QTLs (Supplementary Table 4), which produced additional gestational duration - gene associations and further helped the effort of gene prioritization (see comment 4 from Reviewer #2). Accordingly, we have updated the results and methods sections, as well as the supplementary tables and figures.

6. Based on RNA-based enrichment of reproductive tissues, the authors said that the genetic effects may be timed towards end of gestation (page 8). I find it difficult to understand how this claim is supported by the presented data. Can one exclude the possibility that the loci can have effects on early pregnancy physiological changes to the reproductive tissues, but is less strong to induce miscarriage? A re-visit to this statement or further supportive evidence is warranted.

We agree with the reviewer that such a claim deserves additional supportive evidence. Given our interest in investigating the plausibility that the GWAS of gestational duration is enriched in genes acting late in pregnancy, we decided to take this analysis one step further using stratified LD-score regression. We generated our own annotations of gene sets differentially expressed between labor and non-laboring single cells from myometrium samples (list of genes extracted from Pique-Regi R, JCI Insight, 2022). We confirm an enrichment in genes differentially expressed during labor in a cell type-agnostic analysis (overall enrichment = 1.7, p-value = 7.1×10^{-7} , Extended Data Fig. 1), by using the genes that were differentially expressed in any cell type. This might be relevant in terms of target genes for drug discovery regarding the induction of labor or as tocolytic agents. We did not explore the enrichment at the single-cell level given that different cell types had large differences in the number of differentially expressed genes (from 2 to >3000, Supplementary Fig. 16).

We want to mention as well that this analysis doesn't rule out an effect in early pregnancy (i.e. during decidualization) as has been previously suggested, and we have balanced the interpretation of the results accordingly:

“Previous genetic studies have suggested a critical role of the decidua (endometrium) in the timing of parturition, indicating an effect early in pregnancy. Using stratified LD-score regression, we show that the heritability of gestational duration is enriched in regions harboring genes differentially expressed during labor (enrichment = 1.7, p-value = 7.1×10^{-7} , Extended Data Fig. 1), suggesting the SNPs associated with gestational duration may as well act during labor.”

7. Page 14-15: With COJO analysis, a gestational duration-mediated effect of maternal genetic loci on birth weight has been found using 87 maternal genome SNPs previously found to have suggestive association with birth weight. This is an interesting finding. Can the authors validate the finding showing that maternal

effects on birth weight are partially due to effects on gestational-duration using the sets of ~30 variants with stronger evidence for belonging in the “maternal only effect” cluster by Warrington as well as the more recent study by the deCODE group?

We thank the reviewer for this comment. The reason to use maternal and fetal SNPs with suggestive evidence was to increase the number of SNPs tested, but these could have both maternal and fetal effects. We have now performed the same analysis using summary statistics (multi-trait COJO) and individual level data (using parental transmitted and non-transmitted alleles and genotype dosage) from Norway and Iceland on top variants classified as having a “Maternal Only” or “Fetal Only” effect on birth weight (Warrington N, Nat Genet, 2019). Interestingly, the median reduction in maternal effect on birth weight after conditioning is larger for the genome-wide significant SNPs than when considering those with suggestive evidence (-20.2% vs -11%, respectively), observed using both for individual level data and summary statistics (Supplementary Fig. 14 and Supplementary Table 13). This might be due to the effects of these variants acting almost exclusively through the maternal genome.

We did not assess the effect on variants from the more recent fetal growth study from deCode given that the additional GWAS added in this meta-analysis was adjusted by gestational duration.

Minor comments

1. On page 19, the second paragraph cites Table S14, which should have been Table S13.

We thank the reviewer for noticing this, we have now made sure that all tables/ figures are numbered correctly.

2. In the abstract, the sentence on “...sex-specific...” is not clear unless a reader refers to the Results section. Please rephrase this with a more direct presentation of the finding.

We agree that the sentence highlighted required further context. We have now re-phrased it to represent the results more directly.

3. On page 8 (results), for ease of reading, please add the gestational duration for controls.

We have now added the gestational duration cut-offs for controls.

4. Page 14: Clarify what is meant by 'limited adjustment for gestational duration', whether this context should be considered in the interpretation of the findings/conclusions.

We now provide an improved explanation for "limited adjustment for gestational duration". The GWAS on birth weight we have used was adjusted for gestational duration in < 15% of samples. We have further commented on the implications of this in the discussion section.

5. Figure S3: Add a legend describing the colors of the text (tissues) and dots

We have added a legend describing the colors, both in the figure itself and in the figure caption.

6. Figure S9: Edit the 6th line of the legend

We have now edited line 6 of the legend.

7. Figure S13: Label the plots with names of the three gene regions

This figure is now removed from the manuscript given that we provide the enrichment analysis for all the regions identified.

8. The first paragraph of the introduction ends with a note that sets an expectation that the current work dissected differential genetic effects on clinical subtypes of parturition timing. My expectation was to see

genetic links with subtypes such as PROM, spontaneous onset labor etc, but no such findings were presented. This section and the discussion of this topic in the Discussions section needs tempering or findings of clinical subtypes, if any, added.

We agree with the reviewer that this paragraph and the discussion could lend to expecting clinical subtypes of preterm delivery. We have now modified accordingly to avoid such expectations.

Decision Letter, first revision:

4th October 2022

Dear Pol,

Your revised manuscript "Genetic effects on the timing of parturition and links to fetal birth weight" (NG-A60055R) has been seen by the original referees. As you will see from their comments below, they find that the paper has improved in revision, and therefore we will be happy in principle to publish it in Nature Genetics as an Article pending final revisions to satisfy Reviewer #3's remaining points and to comply with our editorial and formatting guidelines.

We are now performing detailed checks on your paper, and we will send you a checklist detailing our editorial and formatting requirements soon. Please do not upload the final materials or make any revisions until you receive this additional information from us.

Thank you again for your interest in Nature Genetics. Please do not hesitate to contact me if you have any questions.

Sincerely,
Kyle

Kyle Vogan, PhD
Senior Editor
Nature Genetics
<https://orcid.org/0000-0001-9565-9665>

Reviewer #1 (Remarks to the Author):

I appreciate the authors' response and find it reasonable.

Reviewer #2 (Remarks to the Author):

I'm happy with the responses to my points and have no further comments. Congratulations on a very nice manuscript.

Signed: John Perry

Reviewer #3 (Remarks to the Author):

I applaud the authors for addressing my comments and updating the manuscript with several additional analyses, which added further value to this important work.

The evolutionary analysis is tangential and the evidence is not convincingly strong although it goes in line with one of the theories out there. The revisions made in the results following my feedback have toned this down, which is great. I further urge the authors to remove the related phrase "...likely shaped by strong evolutionary forces..." from the conclusion section.

Author Rebuttal, first revision:

Responses to the Reviewers

We thank the reviewers for the constructive comments throughout the revision process and for the encouraging words. In this revision, we have formatted the manuscript according to the author guidelines we were provided, and we have addressed the comment from Reviewer #3.

Reviewer #1:

I appreciate the authors' response and find it reasonable.

Reviewer #2:

I'm happy with the responses to my points and have no further comments. Congratulations on a very nice manuscript.

Signed: John Perry

Reviewer #3:

I applaud the authors for addressing my comments and updating the manuscript with several additional analyses, which added further value to this important work.

The evolutionary analysis is tangential and the evidence is not convincingly strong although it goes in line with one of the theories out there. The revisions made in the results following my feedback have toned this down, which is great. I further urge the authors to remove the related phrase "...likely shaped by strong evolutionary forces..." from the conclusion section.

We agree with the reviewer that removing the mentioned sentence from the conclusion section is pertinent with the evidence reported - we confirm we have done so.

Final Decision Letter:

22nd February 2023

Dear Pol,

I am delighted to say that your manuscript "Genetic effects on the timing of parturition and links to fetal birth weight" has been accepted for publication in an upcoming issue of Nature Genetics.

Due to the importance of these deadlines, we ask that you please let us know now whether you will be difficult to contact over the next month. If this is the case, we ask you provide us with the contact information (email, phone and fax) of someone who will be able to check the proofs on your behalf,

and who will be available to address any last-minute problems.

Your paper will be published online after we receive your corrections and will appear in print in the next available issue. You can find out your date of online publication by contacting the Nature Press Office (press@nature.com) after sending your e-proof corrections. Now is the time to inform your Public Relations or Press Office about your paper, as they might be interested in promoting its publication. This will allow them time to prepare an accurate and satisfactory press release. Include your manuscript tracking number (NG-A60055R1) and the name of the journal, which they will need when they contact our Press Office.

Before your paper is published online, we will be distributing a press release to news organizations worldwide, which may very well include details of your work. We are happy for your institution or funding agency to prepare its own press release, but it must mention the embargo date and Nature Genetics. Our Press Office may contact you closer to the time of publication, but if you or your Press Office have any enquiries in the meantime, please contact press@nature.com.

Please note that Nature Genetics is a Transformative Journal (TJ). Authors may publish their research with us through the traditional subscription access route or make their paper immediately open access through payment of an article-processing charge (APC). Authors will not be required to make a final decision about access to their article until it has been accepted. [Find out more about Transformative Journals](https://www.springernature.com/gp/open-research/transformative-journals)

Authors may need to take specific actions to achieve [compliance](https://www.springernature.com/gp/open-research/funding/policy-compliance-faqs) with funder and institutional open access mandates. If your research is supported by a funder that requires immediate open access (e.g. according to [Plan S principles](https://www.springernature.com/gp/open-research/plan-s-compliance)), then you should select the gold OA route, and we will direct you to the compliant route where possible. For authors selecting the subscription publication route, the journal's standard licensing terms will need to be accepted, including [self-archiving-and-license-to-publish](https://www.nature.com/nature-portfolio/editorial-policies/self-archiving-and-license-to-publish). Those licensing terms will supersede any other terms that the author or any third party may assert apply to any version of the manuscript.

Please note that Nature Portfolio offers an immediate open access option only for papers that were first submitted after 1 January 2021.

If you have not already done so, we invite you to upload the step-by-step protocols used in this manuscript to the Protocols Exchange, part of our on-line web resource, natureprotocols.com. If you complete the upload by the time you receive your manuscript proofs, we can insert links in your article that lead directly to the protocol details. Your protocol will be made freely available upon publication of your paper. By participating in natureprotocols.com, you are enabling researchers to more readily reproduce or adapt the methodology you use. [Natureprotocols.com](http://natureprotocols.com) is fully searchable, providing your protocols and paper with increased utility and visibility. Please submit your protocol to <https://protocolexchange.researchsquare.com/>. After entering your nature.com username and password you will need to enter your manuscript number (NG-A60055R1). Further information can be found at <https://www.nature.com/nature-portfolio/editorial-policies/reporting-standards#protocols>

Sincerely,
Kyle

Kyle Vogan, PhD
Senior Editor
Nature Genetics
<https://orcid.org/0000-0001-9565-9665>